# DifFace: Blind Face Restoration with Diffused Error Contraction

## Abstract

While deep learning-based methods for blind face restoration have achieved unprecedented success, they still suffer from two major limitations. First, most of them deteriorate seriously when facing complex degradations out of their training data. Second, these methods require multiple constraints, e.g., fidelity, perceptual, and adversarial losses, which requires laborious hyper-parameters tuning to stabilize and balance their influences. In this work, we propose a novel method named *DifFace*, being able to cope with unseen and complex degradations more gracefully without complicated loss designs. The key of our method is to establish a posterior distribution from the observed low-quality (LQ) image to its high-quality (HQ) counterpart. In particular, we design a transition distribution from the LQ image to the intermediate state of a pre-trained diffusion model and then gradually transmit from this intermediate state to the HQ target by recursively applying a pre-trained diffusion model. The transition distribution only relies on a restoration backbone that is trained with $L_2$ loss on some synthetic data, which favorably avoids the cumbersome training process in existing methods. Moreover, the transition distribution is capable of contracting the error of the restoration backbone and thus makes our method more robust to unknown degradations. Comprehensive experiments show that *DifFace* is superior to current state-of-the-art methods, especially in cases with severe degradations. Code and model will be released.

## 1 Introduction

Blind face restoration (BFR) aims at recovering a high-quality (HQ) image from its low-quality (LQ) counterpart, which usually suffers from complex degradations, such as noise, blurring, and downsampling. BFR is an extremely ill-posed inverse problem as multiple HQ solutions may exist for any given LQ image.

Approaches for BFR have been dominated by deep learning-based methods (Wang et al., 2021; Tu et al., 2021; Feihong et al., 2022; Gu et al., 2022). The main idea of them is to learn a mapping, usually parametrized as a deep neural network, from the LQ images to the HQ ones based on a large amount of pre-collected LQ/HQ image pairs. In most cases, these image pairs are synthesized by assuming a degradation model that often deviates from the real one. Most existing methods are sensitive to such a deviation and thus suffer a dramatic performance drop when encountering mismatched degradations in real scenarios.

Various constraints have been designed to mitigate the influence of such a deviation and improve the restoration quality. The $L_2$ (or $L_1$) loss is commonly used to ensure fidelity, although these pixel-wise losses are known to favor the prediction of an average (or a median) over the plausible solutions. Recent BFR methods also introduce the adversarial loss (Goodfellow et al., 2014) and the perceptual loss (Johnson et al., 2016; Zhang et al., 2018) to achieve more realistic results. Besides, some existing methods also exploit face-specific priors to further constrain the restored solution, e.g., face landmarks (Chen et al., 2018), facial components (Li et al., 2020), and generative priors (Chan et al., 2022; Pan et al., 2021; Wang et al., 2021; Yang et al., 2021). Considering so many constraints together makes the training unnecessarily complicated, often requiring laborious hyper-parameters tuning to make a trade-off among these constraints. Worse, the notorious instability of adversarial loss makes the training more challenging.

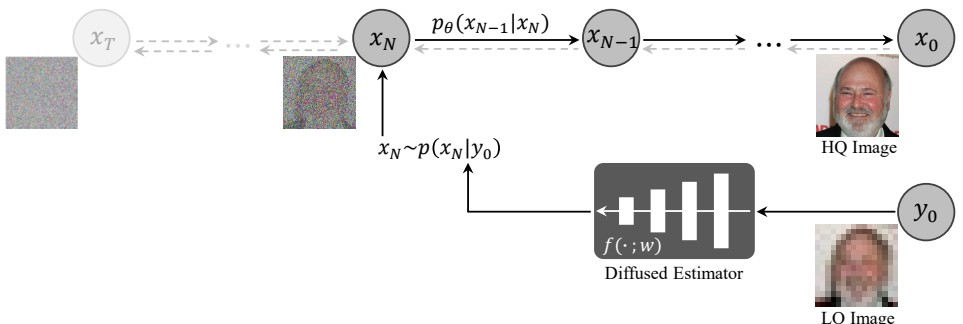

Figure 1: Overview of the proposed method. The solid lines denote the whole inference pipeline of our method. For ease of comparison, we also mark out the forward and reverse processes of the diffusion model by dotted lines.

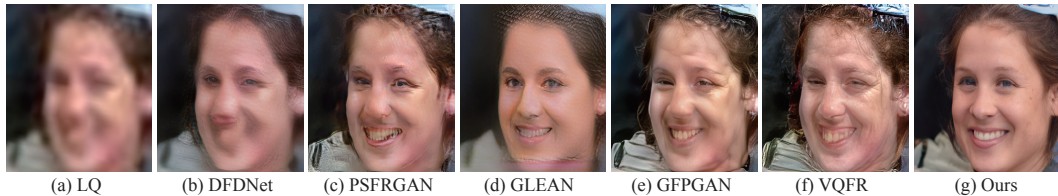

Figure 2: Comparative results of recent state-of-the-art methods and the proposed method on one typical real example. From left to right: (a) low-quality image, (b)-(g) restored results of DFDNet (Li et al., 2020), GLEAN (Chan et al., 2022), PSFRGAN (Chen et al., 2021), GFPGAN (Wang et al., 2021), VQFR (Gu et al., 2022), and our proposed method.

In this work, we devise a novel BFR approach *DifFace*, inspired by recent success of diffusion model (Sohl-Dickstein et al., 2015) in image generation (Dhariwal & Nichol, 2021). Our method does not require multiple constraints in training, and yet is capable of dealing with unknown and complex degradations. Importantly, we leverage the rich image priors and strong generative capability of a pretrained diffusion model without retraining it on any manually assumed degradations.

To this end, we establish a posterior distribution $p(\boldsymbol{x}_0|\boldsymbol{y}_0)$, aiming to infer the HQ image $\boldsymbol{x}_0$ conditioned on its LQ counterpart $\boldsymbol{y}_0$. Due to the complex degradations, solving for this posterior is non-trivial in blind restoration. Our solution to this problem, as depicted in Fig. 1, is to approximate this posterior distribution by a transition distribution $p(\boldsymbol{x}_N|\boldsymbol{y}_0)$, where $\boldsymbol{x}_N$ is a diffused version of the desirable HQ image $\boldsymbol{x}_0$, followed with a reverse Markov chain that estimates $\boldsymbol{x}_0$ from $\boldsymbol{x}_N$. To construct this transition distribution $p(\boldsymbol{x}_N|\boldsymbol{y}_0)$, we introduce a deep neural network which can be simply trained just using $L_2$ loss.

Such a transition distribution is appealing in that it is well motivated by an important observation in DDPM (Ho et al., 2020), where data is destroyed by re-scaling it with a factor of less than 1 and adding noise in the diffusion process. Bringing this notion to our context, the residual between $\boldsymbol{x}_0$ and $\boldsymbol{y}_0$ is also contracted by this factor after diffusion. Our framework uniquely leverages this property by inferring the intermediate diffused variable $\boldsymbol{x}_N$ (where $N < T$) from the LQ image $\boldsymbol{y}_0$, of which the residual to HQ image $\boldsymbol{x}_0$ is reduced. And then from this intermediate state we infer the desirable $\boldsymbol{x}_0$. There are several advantages of doing so: i) our solution is more efficient than the full reverse diffusion process from $\boldsymbol{x}_T$ to $\boldsymbol{x}_0$, ii) we do not need to retrain the diffusion model from scratch, and iii) we can still take advantage of the pre-trained diffusion model via the reverse Markov chain from $\boldsymbol{x}_N$ to $\boldsymbol{x}_0$.

SR3 (Saharia et al., 2022b) also exploits the potentials of diffusion model for blind restoration. It feeds the LQ image into the diffusion model as a condition to guide the restoration at each timestep. This requires one to retrain the diffusion model from scratch on pre-collected training data. Hence, it would still suffer from the issue of degradation mismatch when dealing with real-world data. Different from SR3, our method does not need to train the diffusion model from scratch but sufficiently leverages the prior knowledge contained in the pretrained diffusion model. The unique design on transition distribution $p(\boldsymbol{x}_N|\boldsymbol{y}_0)$ further allows us to cope with unknown degradations.

In summary, the contributions of this work are as follows:

- We devise a new diffusion-based BFR approach to cope with severe and unknown degradations. The notion of formulating the posterior distribution as a Markov chain that starts from the LQ image and ends at the desirable HQ image is novel. We theoretically show that the Markov chain can compress the predicted error by a factor of less than 1.
- We show that the image prior captured in a pretrained diffusion model can be harnessed by having the Markov chain built partially on the reverse diffusion process. Such a unique design also allows us to explicitly control the restoration's fidelity and realism by changing the Markov chain's length.
- We show that BFR can be achieved without complicated losses as in existing methods. Our method only needs to train a neural network with the basic $L_2$ loss, simplifying the training pipeline. A preview of our restored results compared with existing methods are shown in Fig. 2.

## 2   PRELIMINARIES

We provide a brief introduction to the diffusion probabilistic model (Sohl-Dickstein et al., 2015) (known as diffusion model for brevity) to ease the subsequent presentation of our method.

Diffusion model was proposed in Sohl-Dickstein et al. (2015), mainly inspired by the non-equilibrium statistical physics. Ho et al. (2020) built up a novel connection between diffusion model and denoising scoring matching. Furthermore, Song et al. (2020) proposed a unified framework to formulate the diffusion model from the perspective of stochastic differential equation (SDE). Attributed to its sound and perfect theoretical support, diffusion model has achieved impressive success in generation of images (Ho et al., 2020; Dhariwal & Nichol, 2021), audio (Chen et al., 2020), graph (Niu et al., 2020) and shapes (Cai et al., 2020).

Diffusion model consists of a forward process (or diffusion process) and a reverse process. Given a data point $\boldsymbol{x}_0$ with probability distribution $q(\boldsymbol{x}_0)$, the forward process gradually destroys its data structure by repeated application of the following Markov diffusion kernel:

$$q(\boldsymbol{x}_t|\boldsymbol{x}_{t-1}) = \mathcal{N}(\boldsymbol{x}_t; \sqrt{1-\beta_t}\boldsymbol{x}_{t-1}, \beta_t\boldsymbol{I}), \ t = 1, 2, \cdots, T, \tag{1}$$

where $\beta_1, \ldots, \beta_T$ is a pre-defined or learned noise variance schedule. With a rational design on the variance schedule, it theoretically guarantees that $q(\boldsymbol{x}_t)$ converges to the unit spherical Gaussian distribution. It is noteworthy that the marginal distribution at arbitraty timestep $t$ has the following analytical form:

$$q(\boldsymbol{x}_t|\boldsymbol{x}_0) = \mathcal{N}(\boldsymbol{x}_t; \sqrt{\alpha_t}\boldsymbol{x}_0, (1-\alpha_t)\boldsymbol{I}), \tag{2}$$

where $\alpha_t = \prod_{l=1}^{t}(1-\beta_l)$.

As for the reverse process, it aims to learn a transition kernel from $\boldsymbol{x}_t$ to $\boldsymbol{x}_{t-1}$, which is defined as the following Gaussian distribution:

$$p_\theta(\boldsymbol{x}_{t-1}|\boldsymbol{x}_t) = \mathcal{N}\left(\boldsymbol{x}_{t-1}; \boldsymbol{\mu}_\theta(\boldsymbol{x}_t, t), \boldsymbol{\Sigma}_\theta(\boldsymbol{x}_t, t)\right), \tag{3}$$

where $\theta$ is the learnable parameter. With such a learned transition kernel, we can approximate the data distribution $q(\boldsymbol{x}_0)$ via the following marginal distribution:

$$p_\theta(\boldsymbol{x}_0) = \int p(\boldsymbol{x}_T)\prod_{t=1}^{T} p_\theta(\boldsymbol{x}_{t-1}|\boldsymbol{x}_t)\mathrm{d}\boldsymbol{x}_{1:T}, \tag{4}$$

where $p(\boldsymbol{x}_T) = \mathcal{N}(\boldsymbol{x}_T; \boldsymbol{0}, \boldsymbol{I})$.

## 3   PROPOSED METHOD

In this section, we present our BFR method, *DifFace*, and discuss how to exploit the image priors encapsulated in a pretrained diffusion model. To keep the notations consistent with Sec. 2, we denote the LQ image and HQ image as $\boldsymbol{y}_0$ and $\boldsymbol{x}_0$. To restore the desirable HQ image from its degraded counterpart, we need to design a rational posterior distribution of $p(\boldsymbol{x}_0|\boldsymbol{y}_0)$.

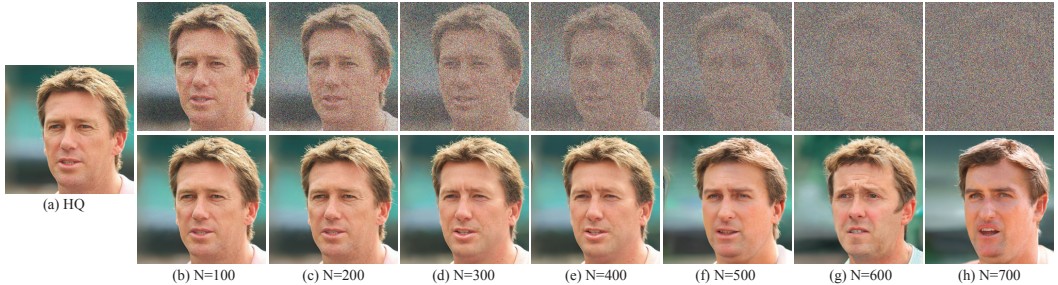

Figure 3: Illustration of the diffused $\boldsymbol{x}_N$ (top row) and the reconstructed results (bottom row) by the pre-trained diffusion model from different starting timesteps. Note that the employed diffusion model is trained with 1000 discrete steps following Dhariwal & Nichol (2021).

## 3.1 MOTIVATION

Considering a diffusion model with $T$ discrete steps, it provides a transition function from $\boldsymbol{x}_t$ to $\boldsymbol{x}_{t-1}$. With the aid of this transition, we can construct the posterior distribution $p(\boldsymbol{x}_0|\boldsymbol{y}_0)$ as follows:

$$p(\boldsymbol{x}_0|\boldsymbol{y}_0) = \int p(\boldsymbol{x}_N|\boldsymbol{y}_0) \prod_{t=1}^{N} p_\theta(\boldsymbol{x}_{t-1}|\boldsymbol{x}_t) \mathrm{d}\boldsymbol{x}_{1:N}, \tag{5}$$

where $1 \leq N < T$ is an arbitraty timestep. To restore $\boldsymbol{x}_0$ from $\boldsymbol{y}_0$, we can leverage the ancestral sampling (Bishop & Nasrabadi, 2006) as follows:

$$\boldsymbol{x}_N \sim p(\boldsymbol{x}_N|\boldsymbol{y}_0), \ \ \boldsymbol{x}_{t-1}|\boldsymbol{x}_t \sim p_\theta(\boldsymbol{x}_{t-1}|\boldsymbol{x}_t), \ \ t = N, N-1, \ldots, 1. \tag{6}$$

Since the transition kernel $p_\theta(\boldsymbol{x}_{t-1}|\boldsymbol{x}_t)$ can be readily borrowed from the pretrained diffusion model, our goal thus turns to design the transition distribution of $p(\boldsymbol{x}_N|\boldsymbol{y}_0)$.

We have an important observation by diving into Eq. (6). If replacing $p(\boldsymbol{x}_N|\boldsymbol{y}_0)$ with the marginal distribution $q(\boldsymbol{x}_N|\boldsymbol{x}_0)$ defined in Eq. (2), it degenerates into the diffusion and reconstruction process for $\boldsymbol{x}_0$ via the pretrained diffusion model, i.e.,

$$\underbrace{\boldsymbol{x}_N \sim q(\boldsymbol{x}_N|\boldsymbol{x}_0)}_{\text{Diffusion}}, \ \ \underbrace{\boldsymbol{x}_{t-1}|\boldsymbol{x}_t \sim p_\theta(\boldsymbol{x}_{t-1}|\boldsymbol{x}_t)}_{\text{Reconstruction}}, \ \ t = N, N-1, \ldots, 1. \tag{7}$$

In Figure 3, we show some diffused and reconstructed results under different settings for the starting timestep $N$. One can observe that when $N$ lies in a reasonable range (e.g., $N < 500$), meaning that $\boldsymbol{x}_0$ is slightly "destroied", it is possible to accurately reconstruct $\boldsymbol{x}_0$ using the pretrained diffusion model.

This observation indicates that $q(\boldsymbol{x}_N|\boldsymbol{x}_0)$ is an ideal choice for the desired $p(\boldsymbol{x}_N|\boldsymbol{y}_0)$ by a reasonable $N$. Since the HQ image $\boldsymbol{x}_0$ is inaccessible in the task of BFR, we thus explore how to design a plausible $p(\boldsymbol{x}_N|\boldsymbol{y}_0)$ to approximate $q(\boldsymbol{x}_N|\boldsymbol{x}_0)$.

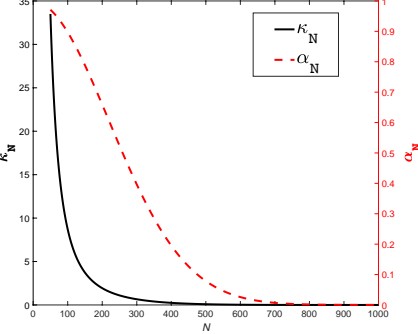

Figure 4: The change curves of $\kappa_N$ and $\alpha_N$ with the starting timestep $N$.

## 3.2 DESIGN

Recall that our goal is to design a transition distribution $p(\boldsymbol{x}_N|\boldsymbol{y}_0)$ to approximate $q(\boldsymbol{x}_N|\boldsymbol{x}_0)$. Fortunately, the target distribution $q(\boldsymbol{x}_N|\boldsymbol{x}_0)$ has an analytical form as shown in Eq. (2). This inspires us to formulate $p(\boldsymbol{x}_N|\boldsymbol{y}_0)$ as a Gaussian distribution as follows:

$$p(\boldsymbol{x}_N|\boldsymbol{y}_0) = \mathcal{N}\left(\boldsymbol{x}_N; \sqrt{\alpha_N} f(\boldsymbol{y}_0; w), (1-\alpha_N)\boldsymbol{I}\right), \tag{8}$$

where $f(\cdot; w)$ is a neural network with parameter $w$, aiming to provide an initial prediction for $\boldsymbol{x}_0$. It should be noted that the final restored result by our method is achieved by sampling from the whole Markov chain of Eq. (6) (see Fig. 1), and not directly predicted by $f(\cdot; w)$. As for $f(\cdot; w)$, it

is only used to construct the marginal distribution of $\boldsymbol{x}_N$, a diffused version of $\boldsymbol{x}_0$, and thus named as "diffused estimator" in this work.

Next we consider the Kullback-Leibler (KL) divergence between the designed $p(\boldsymbol{x}_N|\boldsymbol{y}_0)$ and its target $q(\boldsymbol{x}_N|\boldsymbol{x}_0)$. By denoting the predicted error of $f(\cdot;w)$ as $\boldsymbol{e} = \boldsymbol{x}_0 - f(\boldsymbol{y}_0;w)$, we have

$$D_{\mathrm{KL}}\left[p(\boldsymbol{x}_N|\boldsymbol{y}_0)\|q(\boldsymbol{x}_N|\boldsymbol{x}_0)\right] = \frac{1}{2}\kappa_N\|\boldsymbol{e}\|_2^2, \tag{9}$$

where $\kappa_N = \frac{\alpha_N}{1-\alpha_N}$. As shown in Fig. 4, $\kappa_N$ strictly decreases monotonically with the timestep $N$. Hence, larger $N$ will offer a better approximattion to $q(\boldsymbol{x}_N|\boldsymbol{x}_0)$, and further achieve a more realistic image via the reverse process from $\boldsymbol{x}_N$ to $\boldsymbol{x}_0$ of the pretrained diffusion model. However, $\boldsymbol{x}_N$ will contain more noises when $N$ is getting larger, as shown in Fig. 3. Thus, an overly large $N$ will inevitably deviate the restored result from the ground truth $\boldsymbol{x}_0$. Therefore, the choice of $N$ induces a realism-fidelity trade-off for the restored HQ image. We provide the ablation study in Sec. 4.

### 3.3 DISCUSSION

Instead of directly learning a mapping from $\boldsymbol{y}_0$ to the desirable $\boldsymbol{x}_0$ under multiple constraints like current deep learning-based methods (Chen et al., 2021; Yang et al., 2021; Wang et al., 2021), the proposed method turns to predict $\boldsymbol{x}_N$, a diffused version of $\boldsymbol{x}_0$, from $\boldsymbol{y}_0$. Such a new learning paradigm brings several significant advantages compared to current approaches. Specifically,

- According to the designed $p(\boldsymbol{x}_N|\boldsymbol{y}_0)$ in Eq. (8), we can access the diffused $\boldsymbol{x}_N$ as follows:

$$\boldsymbol{x}_N = \sqrt{\alpha_N}f(\boldsymbol{y}_0;w) + \sqrt{(1-\alpha_N)}\boldsymbol{\zeta} = \sqrt{\alpha_N}\boldsymbol{x}_0 - \sqrt{\alpha_N}\boldsymbol{e} + \sqrt{(1-\alpha_N)}\boldsymbol{\zeta}, \tag{10}$$

 where $\boldsymbol{e} = \boldsymbol{x}_0 - f(\boldsymbol{y}_0;w)$, $\boldsymbol{\zeta} \sim \mathcal{N}(\boldsymbol{\zeta}|\boldsymbol{0},\boldsymbol{I})$. It can be seen that the predicted error $\boldsymbol{e}$ is contracted by a factor of $\sqrt{\alpha_N}$ that is less than 1 as shown in Fig. 4.

 Attributed to such an error contracttion, our method has a greater error tolerance to the diffused estimator $f(\cdot;w)$. Hence, $f(\cdot;w)$ *can be simply trained with $L_2$ loss on some synthetic data in this work*. This unique flexibility circumvents the sophisticated training process compared with most recent methods (Li et al., 2020; Wang et al., 2021; Gu et al., 2022). Our experiments (Sec. 5) show that the proposed method consistently obtains satisfactory restoration results over existing methods, especially when dealing with severe and complex degradations.

- After obtaining the diffused $\boldsymbol{x}_N$ via Eq. (10), our method gradually generates the desirable HQ result by sampling recursively from $p_\theta(\boldsymbol{x}_{t-1}|\boldsymbol{x}_t)$ starting from $t = N$ and ending at $t = 1$. Through this sampling procedure, we effectively leverage the rich image priors and powerful generation capability of the pretrained diffusion model to help the restoration task. Unlike existing methods, since the diffusion model is completely trained on the HQ images in an unsupervised manner, it thus reduces the dependence of our method on the manually synthesized training data, of which the distribution may deviate from the true degradation.

In summary, our designed posterior distribution $p(\boldsymbol{x}_0|\boldsymbol{y}_0)$ in Eq. (5) renders a more robust learning paradigm for BFR.

## 4 MODEL ANALYSIS

In this section, we analyze the influence of the starting timestep $N$ and the diffused estimator $f(\cdot;w)$ to our model. We consider two classical network architectures as the backbone for $f(\cdot;w)$, i.e., SRCNN (Dong et al., 2015) and SwinIR (Liang et al., 2021).

**Training Settings.** We train the diffused estimator $f(\cdot;w)$ on the FFHQ dataset (Karras et al., 2019) that contains 70000 HQ face images. We firstly resize the HQ images into a resolution of $512 \times 512$, and then synthesize the LQ images following a typical degradation model used in recent literature (Wang et al., 2021):

$$\boldsymbol{y} = \left\{[(\boldsymbol{x} * \boldsymbol{k}_l) \downarrow_s + \boldsymbol{n}_\sigma]_{\mathrm{JPEG}_q}\right\} \uparrow_s, \tag{11}$$

where $\boldsymbol{y}$ and $\boldsymbol{x}$ are the LQ and HQ image, $\boldsymbol{k}_l$ is the Gaussian kernel with kernel width $l$, $\boldsymbol{n}_\sigma$ is Gaussian noise with standard deviation $\sigma$, $*$ is 2D convolutional operator, $\downarrow_s$ and $\uparrow_s$ are the Bicubic

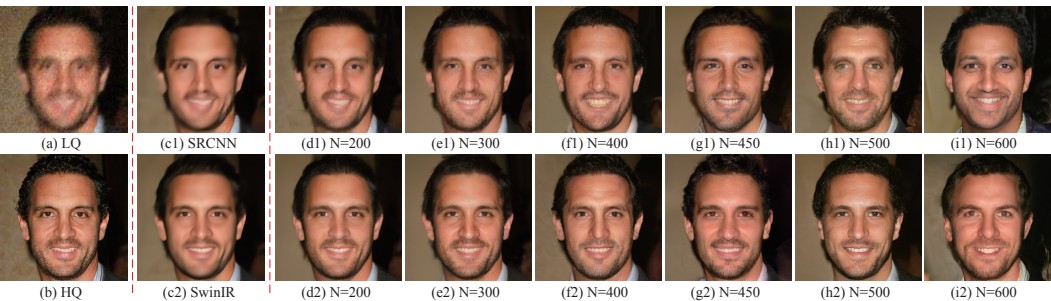

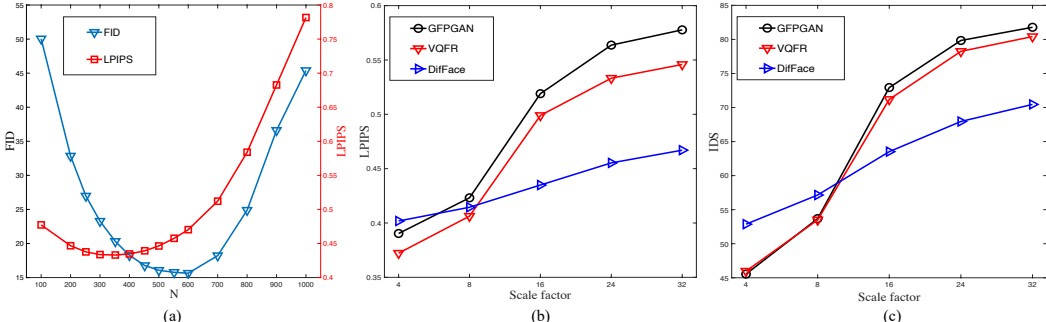

Figure 5: An example restored by *DifFace* under different settings of the starting timestep $N$ and the diffused estimator $f(\cdot; w)$. (a) LQ image, (b) HQ image, (c1)-(c2) restored results by SRCNN and SwinIR, (d1)-(i1) restored results by *DifFace* that takes SRCNN as diffused estimator, (d2)-(i2) restored results by *DifFace* that takes SwinIR as diffused estimator.

Figure 6: From left to right: (a) FID and LPIPS with respect to the starting timestep $N$ on the validation dataset, (b)-(c) LPIPS and IDS with respect to the downsampling scale factor (i.e., $s$ in Eq. (11)) on the testing dataset.

downsampling or upsampling operators with scale $s$, and $[\cdot]_{\text{JPEG}_q}$ represents the JPEG compression process with quality factor $q$. And the hyper-parameters $l$, $s$, $\sigma$, and $q$ are uniformly sampled from $[0.1, 15]$, $[0.8, 32]$, $[0, 20]$, and $[30, 100]$ respectively. To evaluate the performances of *DifFace* under different settings for $N$ and $f(\cdot; w)$, we selected 3500 HQ images from CelebA-HQ (Karras et al., 2018) and generated the LQ images via Eq. (11) as our validation dataset.

We adopt the Adam (Kingma & Ba, 2015) algorithm to optimize the network parameters $w$ under $L_2$ loss. The batch size is set as 16, and other settings of Adam follow the default configurations of Pytorch (Paszke et al., 2019). We train the model for 600k iterations, and the learning rate is decayed gradually from $1e$-4 to $1e$-6 with the cosine annealing schedule (Loshchilov & Hutter, 2017). As for the diffusion model, it contains 1000 discrete steps with linear noise schedule, and is trained on FFHQ (Karras et al., 2018) dataset based on the official code[1] of Dhariwal & Nichol (2021).

**Starting Timestep.** In Fig. 5, we show some restored results by *DifFace* under different settings for the starting timesteps $N$. More real-world visual examples can be found in Fig. 10 of Appendix. The results show that one can make a trade-off between realism and fidelity through different choices of $N$. In particular, if we set a larger $N$, the restored results would appear more realistic but with lower fidelity in comparison to the ground truth HQ images. The phenomenon is reflected by the average FID (Heusel et al., 2017) and LPIPS (Zhang et al., 2018) scores with respect to $N$ in Fig. 6 (a). The proposed *DifFace* performs very well in the range of $[400, 450]$, and we thus set $N$ as 400 throughout the whole experiments in this work. In practice, we speedup the inference process four times following Nichol & Dhariwal (2021), and thus sample 100 steps for each testing image.

**Diffused Estimator.** Figure 5 displays an example restored by *DifFace*, which either takes SR-CNN (Dong et al., 2015) or SwinIR (Liang et al., 2021) as the diffused estimator $f(\cdot; w)$. More details and comparisons on these two backbones are provided in Appendix C. Even with the simplest SRCNN that only contains several plain convolutional layers, *DifFace* is able to restore a plausible

---

[1] https://github.com/openai/guided-diffusion

Table 1: The quantitative comparisons of different methods on CelebA-Test. The best and second best results are highlighted in **bold** and underline, respectively.

| Metrics | Input | Methods | | | | | | |
|---|---|---|---|---|---|---|---|---|
| | | DFDNet | PULSE | PSFRGAN | GLEAN | GFPGAN | VQFR | *DifFace* |
| PSNR↑ | 23.41 | 23.15 | 22.14 | 22.74 | 23.41 | 22.18 | 21.94 | **24.08** |
| SSIM↑ | 0.702 | 0.629 | 0.682 | 0.630 | 0.666 | 0.631 | 0.585 | **0.703** |
| LPIPS↓ | 0.604 | 0.554 | 0.508 | 0.500 | 0.469 | 0.495 | 0.471 | **0.435** |
| FID↓ | 283.48 | 64.65 | 48.27 | 52.08 | 59.84 | 26.90 | 25.74 | **18.20** |
| IDS↓ | 78.32 | 86.21 | 74.97 | 68.14 | 67.13 | 66.76 | 65.87 | **62.39** |

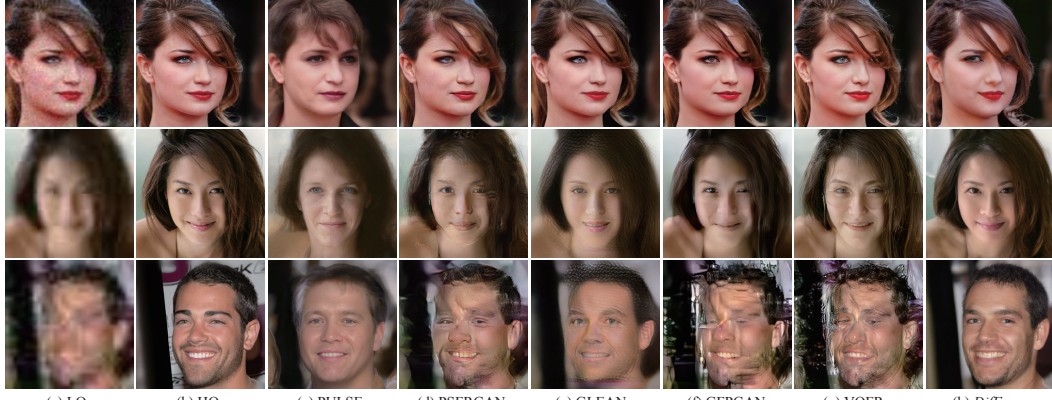

(a) LQ     (b) HQ     (c) PULSE     (d) PSFRGAN     (e) GLEAN     (f) GFPGAN     (g) VQFR     (h) *DifFace*

Figure 7: Visual comparisons of different methods on the synthetic dataset CelebA-Test.

HQ image under severe degradations. Using a more elaborated architecture like SwinIR results in more apparent details (e.g., hairs). The results suggest the versatility of *DifFace* in the choices of the diffused estimators. In the following experiments, we use SwinIR as our diffused estimator for $f(\cdot; w)$.

## 5 EXPERIMENTAL RESULTS

In this section, we conducted extensive experiments to verify the effectiveness of *DifFace* on both synthetic and real-world datasets. The training follows the settings detailed in Sec. 4.

### 5.1 EVALUATIONAL SETUP

**Testing Datasets.** We evaluate *DifFace* on one synthetic dataset and three real-world datasets. The synthetic dataset, denoted as CelebA-Test, contains 3500 HQ images from CelebA-HQ (Karras et al., 2018), and the corresponding LQ images are synthesized via Eq. (11). The specific settings on the hyper-parameters in the degradation can be found in Appendix B. As for the real-world datasets, we consider three typical ones with different degrees of degradation, namely LFW-Test, WebPhoto-Test (Wang et al., 2021), and WIDER-Test (Zhou et al., 2022). LFW-Test consists of 1711 mildly degraded face images in the wild, which contains one image for each person in LFW dataset (Huang et al., 2008). WebPhoto-Test is made up of 407 face images crawled from the internet. Some of them are old photos with severe degradations. WIDER-Test selects 970 face images with very heavy degradations from the WIDER Face dataset (Yang et al., 2016), and is thus suitable to test the robustness of different methods under severe degradations.

**Comparison Methods.** We compare *DifFace* with six recent BFR methods, including DFDNet (Li et al., 2020), PULSE (Menon et al., 2020), PSFRGAN (Chen et al., 2021), GLEAN (Chan et al., 2022), GFPGAN (Wang et al., 2021), and VQFR (Gu et al., 2022).

### 5.2 EVALUATION ON SYNTHETIC DATA

To evaluate different methods comprehensively, we adopt five quantitative metrics, namely PSNR, SSIM (Wang et al., 2004), LPIPS (Zhang et al., 2018), FID (Heusel et al., 2017), and Identity Score

Table 2: FID scores of different methods on three real-world testing datasets. The best and second best results are highlighted in **bold** and underline, respectively.

| Datasets | # Images | Methods | | | | | | |
|---|---|---|---|---|---|---|---|---|
| | | DFDNet | PULSE | PSFRGAN | GLEAN | GFPGAN | VQFR | *DifFace* |
| WIDER-Test | 970 | 58.23 | 69.55 | 49.85 | 70.12 | 39.76 | 44.54 | **37.49** |
| LFW-Test | 1711 | 59.83 | 65.17 | 49.80 | 46.39 | 50.13 | 50.88 | **45.23** |
| WebPhoto-Test | 407 | 92.82 | 86.40 | 85.45 | 94.34 | 87.86 | **75.46** | 85.52 |

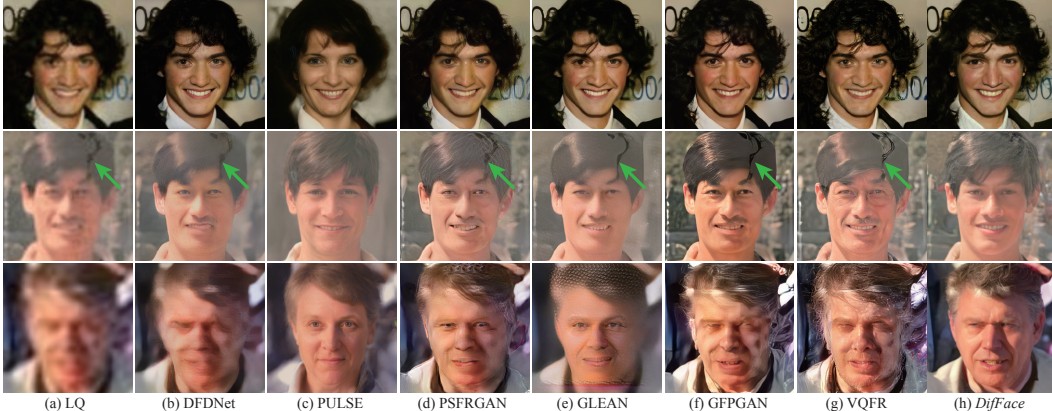

(a) LQ    (b) DFDNet    (c) PULSE    (d) PSFRGAN    (e) GLEAN    (f) GFPGAN    (g) VQFR    (h) *DifFace*

Figure 8: Visual comparisons of different methods on the real-world examples from LFW-Test (first row), WebPhoto-Test (second row), and WIDER-Test (third row).

(IDS). Specifically, LPIPS is a learned perceptual similarity metric calculated based on deep features of VGG (Simonyan & Zisserman, 2015) network. IDS is the embedding angle of ArcFace (Deng et al., 2019) between two images, and mainly reflects identity preservation. As for FID, it is the KL divergence between the feature distributions (assumed as Gaussian) of the restored images and the ground truth images.

We summarize the comparative results on CelebA-Test in Table 1. The proposed *DifFace* achieves the best performance across all five metrics, indicating its effectiveness and superiority in the task of BFR. To further verify the robustness of *DifFace*, we compare it with GFPGAN and VQFR using LPIPS and IDS under different degrees of degradation in Fig. 6(b) and (c). We generate different degrees of degradation by introducing five downsampling scale factors, i.e., 4, 8, 16, 24, and 32, to the images in CelebA-Test. A total of 700 image pairs are generated for each scale. Figure 6(b) and (c) record the averaged performance of different methods over these 700 images with respect to the five downsampling scale factors. While *DifFace* is slightly inferior to VQFR and GFPGAN under the scale factors 4 and 8 (mild degradations), it surpasses other methods and its performance drops more gracefully in the cases of larger factors (severe degradations). This results substantiate the robustness of our *DifFace* in scenarios with very severe degradations. The outcome is consistent with the analysis in Sec. 3.3.

For easy visulization, three typical examples of the CelebA-Test are shown in Fig. 7. In the first example with mild degradation, all methods except PULSE are able to restore a realistic-looking image. PULSE fails to preserve the identity because its optimization cannot find the correct latent code in GAN through inversion. In the second and third examples that exhibit more severe degradation, most compared methods produce obvious artifacts. PULSE returns a plausible face but fails to preserve the identity. Under this challenging degradation setting, *DifFace* still achieves pleasant results. Such robustness complies with our observation in Fig. 6(b) and (c). For more quantitative results, please refer to Fig. 15 in Appendix.

## 5.3 EVALUATION ON REAL-WORLD DATA

In the experiments on real-world datasets, we mainly adopt FID as the quantitative metric since the ground truths are inaccessible. We firstly estimate the feature statistics on the restored images and the HQ images in FFHQ (Karras et al., 2019) dataset, respectively, and then calculate the KL divergence as FID. We attempted other non-reference metrics, namely NIQE (Mittal et al., 2012), NRQM (Ma

Figure 9: Unlike most existing BFR methods, *DifFace* can generate multiple diverse and plausible results given a LQ image, by setting different random seeds for the diffusion model. This example is extracted from real-world dataset WIDER-Test.

Table 3: Comparisons of different methods in terms of model size (M) and runtime (s). For *DifFace*, "A+B" denotes the number of parameters in $f(\cdot; w)$ (A) and the pre-trained diffusion model (B).

| Methods | DFDNet | PULSE | PSFRGAN | GLEAN | GFPGAN | VQFR | *DifFace* |
|---|---|---|---|---|---|---|---|
| Running Time (s) | 1.404 | 3.928 | 0.062 | 0.176 | 0.039 | 0.243 | 4.32 |
| # Parameters (M) | 240.11 | 24.11 | 67.26 | 210.84 | 76.21 | 76.56 | 15.79+159.59 |

et al., 2017), and PI (Blau et al., 2018). These metrics are not adopt here as we empirically found that they behave unusually pessimistic for diffusion-based methods. For completeness, we provide the experimental results on non-reference metrics in Appendix F.

The comparative results are summarized in Table 2. We can observe that *DifFace* achieves the best performance on both WIDER-Test and LFW-Test. On the WebPhoto-Test, it also surpasses most of recent BFR methods except VQFR and PSFRGAN. It should be noted that the FID values on WebPhoto-Test may not be representative as the dataset contains too few images (totally 407) to estimate the feature distribution of the restored images. To supplement the analysis, we show three typical examples of these datasets in Fig. 8, and more visual results are given in Fig. 16 of Appendix. It is again observed that *DifFace* provides better restored results, especially in the second and third examples with severe unknown degradations.

Most existing BFR methods produce only one result for each LQ input, although there may be many reasonable possibilities. This is because they only learn a deterministic mapping between the LQ and HQ images. It is interesting to note that *DifFace*, as shown in Fig. 9, is capable of producing multiple diverse and plausible HQ solutions for any given LQ image by setting different random seeds for the pretrained diffusion model. More results are shown in Fig. 17 and 18 in Appendix. This "pluralistic" property is favorable in BFR, as there exist many different HQ images that can generate the same LQ counterpart.

## 5.4 LIMITATION

Despite the good performance of *DifFace*, the inference efficiency of our method is limited by the iterative sampling process in the diffusion model. Table 3 lists the comparison of different methods on model size, i.e., number of learnable parameters, and running time. It is observed that the model size of *DifFace* (inclusive the pretrained diffusion model) is comparable to that of GLEAN and DFDNet. As for the runing time, it takes 4.32s to deal with a testing example under current setting (100 sampling steps for inference), which is slower than most of competing methods. It is possible to further reduce the sampling process to 20 steps without evident performance drop, and the detailed performance comparisons on different acceleration settings are analyzed in Appendix G. This cuts the running time to 0.92s, lying between DFDNet and VQFR.

## 6 CONCLUSION

We have proposed a new BFR method called *DifFace* in this work. *DifFace* is appealing as it only relies on a restoration backbone that is trained with $L_2$ loss. This vastly simplifies the complicated training objectives in most current approaches. Importantly, we have proposed a posterior distribution that is well-suited for BFR. It consists of a transition kernel and a Markov chain partially borrowed from a pre-trained diffusion model. The former acts as a error compressor, and thus makes our method more robust to severe degradations. The latter effectively leverages the powerful diffusion model to facilitate BFR. Extensive experiments have demonstrated the effectiveness and robustness of our method both on synthetic and real-world datasets. We hope that this work could inspire more robust diffusion-based restoration methods in the future.

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

## A  RELATED WORK

We provide a more detailed discussion here on related work. The research on BFR mainly focused on exploring better face priors to obtain more rational results. In the following, we review a few popular priors for BFR, namely geometric priors, reference priors, and generative priors. Diffusion prior can be considered as a type of generative prior.

**Geometric Priors**. Face images are highly structured compared with natural images. The structural information, such as facial landmarks (Chen et al., 2018), face parsing maps (Chen et al., 2021; Shen et al., 2018; Chen et al., 2021), facial component heatmaps (Li et al., 2020), and 3D shapes (Ren et al., 2019; Hu et al., 2020; Zhu et al., 2022), can be used a guidance to facilitate the restoration process. The geometric face priors estimated from degraded inputs can be unreliable, affecting the performance of the subsequent BFR task.

**Reference Priors**. Some existing methods (Li et al., 2018; Dogan et al., 2019) guide the restoration with an additional HQ reference image that owns the same identity as the degraded input. The main limitations of these methods stem from their dependence on the HQ reference images, which are inaccessible in some scenarios. Li et al. (2020) address this limitation by constructing an offline facial component dictionary based on the features extracted from HQ images. Then, it searches for the closest facial components in this dictionary for the given LQ images during restoration.

**Generative Priors**. Unlike Li et al. (2020), more recent approaches directly exploit the rich priors encapsulated in generative models for BFR. Following the paradigm of GAN inversion (Xia et al., 2022), the earliest explorations (Menon et al., 2020; Gu et al., 2020; Pan et al., 2021) iteratively optimize the latent code of a pre-trained GAN for the desirable HQ target. To circumvent the time-consuming optimization, some studies (Chan et al., 2022; Wang et al., 2021; Yang et al., 2021) directly embed the decoder of the pre-trained StyleGAN (Karras et al., 2019) into the BFR network, and evidently improve the restoration performance. The success of VQGAN (Esser et al., 2021) in image generation also inspires several BFR methods. These methods mainly design different strategies, e.g., cross-attention (Wang et al., 2022) and parallel decoder (Gu et al., 2022), to improve the matching between the codebook elements of the degraded input and the underlying HQ image. Even though these methods have performed well in most scenarios, they still failed to handle some examples with complicated degradations out of their training data.

Attributed to the powerful capability of diffusion model in image generation, a flurry of works based on diffusion model have been proposed recently. Typically, SR3 (Saharia et al., 2022b) and SRDiff (Li et al., 2022) both feed the observed LQ image into the diffusion model as a condition to guide the restoration in training. Following a similar modeling manner, Rombach et al. (2022) proposed to train the diffusion model in the latent space to accelerate the inference speed. These methods all require one to retrain the diffusion model from scratch on some pre-collected data. The learned model would be still very sensitive to the degradation mismatch when generalizing to other datasets. Therefore, this paper devises a new learning paradigm based on a pretrianed diffusion model to address these issues.

## B  SETUP ON VALIDATION AND TESTING DATASETS

In the main text, we synthesized one validation dataset and one testing dataset, each of which contains 3,500 HQ images randomly selected from CelebA-HQ (Karras et al., 2018). The former is used to select a reasonable starting timestep $N$ for *DifFace*, and the latter to evaluate the performance of different methods. To synthesize the LQ images, we employ the degradation in Eq. (11) with the following settings:

$$s \in \{4, 8, 16, 24, 32\}, \ \sigma \in \{1, 5, 10, 15, 20\}, \ q \in \{30, 40, 50, 60, 70\},$$

$$l_x, l_y \in \{2, 4, 6, 8, 10, 12, 14\}, \ \theta \in \{0, \frac{1}{4}\pi, \frac{1}{2}\pi, \frac{3}{4}\pi\},$$

where $l_x$, $l_y$, and $\theta$ control the generation of the blur kernel $\boldsymbol{k}$. Specifically, the covariance matrix $\boldsymbol{\Sigma}$ of $\boldsymbol{k}$ is defined as follows:

$$\boldsymbol{U} = \begin{bmatrix} \cos\theta & -\sin\theta \\ \sin\theta & \cos\theta \end{bmatrix}, \ \boldsymbol{\Lambda} = \begin{bmatrix} l_x^2 & 0 \\ 0 & l_y^2 \end{bmatrix}, \ \boldsymbol{\Sigma} = \boldsymbol{U}\boldsymbol{\Lambda}\boldsymbol{U}^T. \tag{12}$$

Table 4: Quantitative comparisons of SRCNN, SwinIR and *DifFace* on CelebA-Test. "*DifFace*(X)" means that it takes the backbone "X" as the diffused estimator $f(\cdot; w)$. The parameters in "*DifFace*(X)" includes both of that in the backbone "X" and the pre-trained diffusion model (159.59M).

| Methods | Metrics | | | | | |
|---|---|---|---|---|---|---|
| | PSNR↑ | SSIM↑ | LPIPS↓ | FID↓ | IDS↓ | # Parameters(M) |
| SRCNN | 24.72 | 0.736 | 0.528 | 102.89 | 67.39 | 1.03 |
| *DifFace*(SRCNN) | 23.72 | 0.695 | 0.461 | 25.37 | 65.78 | 1.03+159.59 |
| SwinIR | 25.48 | 0.749 | 0.482 | 87.17 | 58.26 | 15.79 |
| *DifFace*(SwinIR) | 24.08 | 0.703 | 0.435 | 18.20 | 62.39 | 15.79+159.59 |

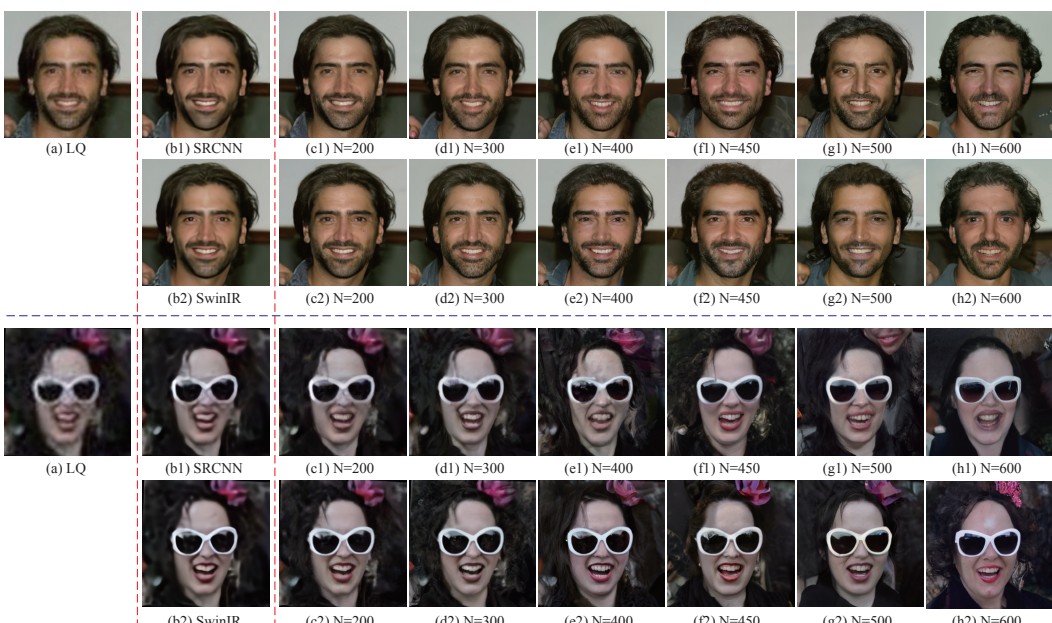

Figure 10: Qualitative comparisons of *DifFace* on LFW-Test (top row) and WIDER-Test (bottom row) under different settings for the starting timestep $N$ and the diffused estimator $f(\cdot; w)$. (a) LQ image, (b1)-(b2) restored results by SRCNN and SwinIR, (c1)-(h1) restored results by *DifFace* that takes SRCNN as diffused estimator, (c2)-(h2) restored results by *DifFace* that takes SwinIR as diffused estimator.

## C DIFFUSED ESTIMATOR

### C.1 NETWORK ARCHITECTURES

We consider two typical restoration backbones for the diffused estimator $f(\cdot; w)$, namely SR-CNN (Dong et al., 2015) and SwinIR (Liang et al., 2021). To apply them in our method, we slightly adjust their settings. The LQ/HQ image pairs in our method are both with size $512 \times 512$, we add two (or three) PixelUnshuffle (Shi et al., 2016) layers with a downscale factor 2 to reduce the input size to $128 \times 128$ (or $64 \times 64$) for SRCNN (or SwinIR). After each PixelUnshuffle layers except the last one, one convolutional and LeakyReLU layers are followed to fuse the features. Similarly, two (or three) PixelShuffle (Shi et al., 2016) layers are also added to the tail of SRCNN (or SwinIR) to upsample the size back to $512 \times 512$.

As for SRCNN, we adopt nine convolutional layers with kernel size 5 between the PixelUnshuffle and PixelShuffle layers, and each convolutional layer has 64 channels. As for SwinIR, we follow the official settings[2] for real-world image super-resolution task with a scale factor 8.

---

[2]https://github.com/JingyunLiang/SwinIR

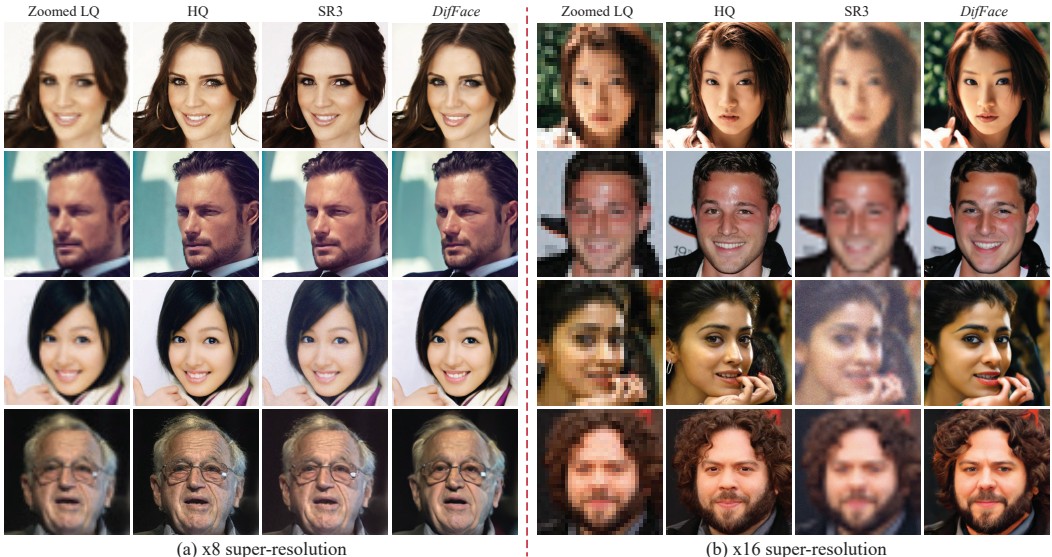

|  Zoomed LQ | HQ | SR3 | *DifFace* | Zoomed LQ | HQ | SR3 | *DifFace* |

(a) x8 super-resolution | (b) x16 super-resolution

Figure 11: Visual comparisons of SR3 and *DifFace* on 8x (left) and 16x (right) face image super-resolution.

## C.2 PERFORMANCE COMPARISON

To quantitatively evaluate these two restoration backbones and their influences to *DifFace*, we conduct experiments on CelebA-Test and summarize the results in Table 4. We also provide some quantitative comparisons in Figure 10 on the real datasets LFW-Test (top row) and WIDER-Test (bottom row). The results offer a few observations as follows:

- Even though taking the simplest backbone SRCNN as our diffused estimator, *DifFace* is still able to produce highly natural-looking results with realistic image details, such as hairs and stubbles. This delivers that the proposed *DifFace* does not rely on specific designs on the diffused estimator.

- Owning to more advanced network architectures as well as more learnable parameters, SwinIR outperforms SRCNN in both qualitative and quantitative comparisons.

- When the starting timestep $N$ gets larger, *DifFace* is capable of achieving more realistic results. However, larger $N$ will introduce more randomness and thus reduce the fidelity.

Based on these observations, we suggest taking SwinIR as the diffused estimator attributed to its superior performance. As for the starting timestep $N$, the proposed *DifFace* performs stably and well when fixing it in the range $[400, 450]$.

## D BICUBIC SUPER-RESOLUTION

Diffusion model can be used for image restoration by directly introducing the LQ image $\boldsymbol{y}_0$ as a condition to guide the generation of $\boldsymbol{x}_0$ (Saharia et al., 2022a;b; Song et al., 2020), namely

$$p(\boldsymbol{x}_0|\boldsymbol{y}_0) = \int p(\boldsymbol{x}_T) \prod_{t=1}^{T} p_\theta(\boldsymbol{x}_{t-1}|\boldsymbol{x}_t, \boldsymbol{y}_0) \mathrm{d}\boldsymbol{x}_{1:T}, \tag{13}$$

where $p(\boldsymbol{x}_T) = N(\boldsymbol{x}_T|\boldsymbol{0}, \boldsymbol{I})$, $p_\theta(\boldsymbol{x}_{t-1}|\boldsymbol{x}_t, \boldsymbol{y}_0)$ is the learned transition kernel from $\boldsymbol{x}_t$ to $\boldsymbol{x}_{t-1}$ conditioned on $\boldsymbol{y}_0$. To access such a transition kernel, an intuitive way is to retrain the diffusion model from scratch. Specifically, in each timestep $t$, $\boldsymbol{x}_t$ and $\boldsymbol{y}_0$ are concatenated together as input to predict $\boldsymbol{x}_{t-1}$ in training. Following this paradigm, SR3 (Saharia et al., 2022b) trained a diffusion model for image super-resolution based on bicubic degradation.

We omit the comparison against SR3 (Saharia et al., 2022b) in the main text as it can only deal with bicubic degradation. Here, we extend the proposed *DifFace* to bicubic super-resolution and make a

Table 5: Quantitative results of SR3 and *DifFace* on bicubic face image super-resolution.

| Methods | Scale factors | Metrics | | | |
|---|---|---|---|---|---|
| | | PSNR↑ | SSIM↑ | LPIPS↓ | IDS↓ |
| SR3 | x8 | 22.74 | 0.585 | 0.505 | 25.66 |
| *DifFace* | | 27.24 | 0.754 | 0.375 | 38.89 |
| SR3 | x16 | 20.50 | 0.506 | 0.645 | 64.19 |
| *DifFace* | | 25.03 | 0.712 | 0.412 | 51.36 |

Table 6: Non-reference metrics of different methods on the real-world dataset WIDER-Test. "Diffusion" denotes the average results on 3,000 images randomly generated by the pretrained diffusion model, which can be regarded as the upper bound of *DifFace*.

| Metrics | Methods | | | | | | | |
|---|---|---|---|---|---|---|---|---|
| | DFDNet | PULSE | PSFRGAN | GLEAN | GFPGAN | VQFR | *DifFace* | Diffusion |
| NIQE↓ | 5.67 | 5.27 | 3.89 | 5.13 | 3.81 | 3.02 | 4.24 | 4.11 |
| NRQM↑ | 6.80 | 4.05 | 8.01 | 7.03 | 8.07 | 8.78 | 6.11 | 6.60 |
| PI↓ | 4.81 | 5.88 | 3.33 | 4.45 | 3.08 | 2.17 | 4.46 | 4.17 |

fair comparison with SR3. We do not make any changes apart from upsampling an LQ image with bicubic interpolation and feeding it into *DifFace* to restore the high-resolution target. Although the degradation model we use (Eq. (11)) does not contain the bicubic degradation, we do not retrain or finetune our diffused estimator on such degradation.

Since the code of SR3 is not released, we thus adopt an unofficially re-implemented version[3] instead. This model is specifically trained for 8x face image super-resolution from size $64 \times 64$ to $512 \times 512$. To further evaluate the generalization capability to different degradation, we also test its performance on the task of 16x super-resolution from $32 \times 32$ to $512 \times 512$. As for the testing dataset, we randomly select 200 images from CelebA-HQ (Karras et al., 2018).

Table 5 lists the quantitative results of SR3 and *DifFace* on the task of bicubic super-resolution, and the corresponding visual comparisons are shown in Fig. 11. *DifFace* achieves better or at least comparable performance on 8x super-resolution, even though SR3 is specifically trained for this bicubic upsampling task. When generalized to 16x super-resolution, *DifFace* outperforms SR3 on both qualitative and qualitative comparisons, indicating the robustness of *DifFace* to unknown degradations. On the other hand, *DifFace* is more efficient than SR3, because SR3 has to pass through the whole reverse process of the diffusion model while *DifFace* starts from the intermediate state (i.e., $x_N$) of the reverse Markov chain.

Another popular diffusion based image super-resolution model is LDM (Rombach et al., 2022). It trains a diffusion model to approximate the conditional distribution in Eq. (13) in the latent space of VQGAN (Esser et al., 2021), which evidently accelerates its inference speed. Since LDM only releases a pretrained model trained on natural images, we thus cannot provide comparisons with it on the task of BFR.

## E  ATMOSPHERIC TURBULENCE RESTORATION

In this section, we consider a more challenging degradation type, namely atmospheric turbulence. It causes geometric distortion and blur effects on the captured images, and negatively affects the face-related downstreaming tasks. Note that none of the *DifFace* and the other comparison methods contains any atmospheric turbulence or similar degradation types. Hence, it is suitable to compare the robustness of different methods on such a degradation type.

It is difficult to collect a large real-world dataset degraded by atmospheric turbulence, since most of the relevant datasets are privately protected due to their military use. We cropped eight real-world image pairs from the manuscript of Mei & Patel (2021), and resize them to the resolution of 512. Each pair contains one image degraded by atmospheric turbulence and one corresponding reference image. We call this dataset as Turb8-Test in this work.

---

[3] https://github.com/Janspiry/Image-Super-Resolution-via-Iterative-Refinement

Table 7: Performance comparisons of DFDNet, VQFR, and *DifFace* under different acceleration settings. "*Dif*(A/B)" means that the whole reverse process of the pretrained diffusion model contains A sampling steps after acceleration, and the starting timestep $N$ in *DifFace* is set as B. In the main text of this paper, we reported the performance of *Dif*(250/100) as our result.

| Metrics | Methods | | | | | | |
|---|---|---|---|---|---|---|---|
| | *Dif*(1000/400) | *Dif*(250/100) | *Dif*(100/40) | *Dif*(50/20) | *Dif*(20/8) | DFDNet | VQFR |
| PSNR↑ | 24.01 | 24.08 | 24.09 | 24.00 | 22.87 | 23.15 | 21.94 |
| SSIM↑ | 0.692 | 0.703 | 0.707 | 0.711 | 0.673 | 0.629 | 0.585 |
| LPIPS↓ | 0.441 | 0.434 | 0.435 | 0.440 | 0.497 | 0.554 | 0.471 |
| FID↓ | 17.51 | 18.20 | 21.00 | 24.45 | 26.97 | 64.65 | 25.74 |
| IDS↓ | 62.56 | 62.39 | 62.08 | 61.77 | 60.95 | 86.21 | 65.87 |
| Runtime↓ | 17.09 | 4.32 | 1.77 | 0.92 | 0.41 | 1.40 | 0.24 |

Figure 12 shows the qualitative comparison results of different methods on Turb8-Test dataset. *DifFace* is able to achieve photo-realistic results and evidently outperforms other methods in visual quality. In particular, the first three examples represent the same person but with large pose variations, *DifFace* still performs well in this scenario. Unfortunately, *DifFace* sometimes cannot preserve the identity very well when the input loses too much image information, e.g., the last example in Fig. 12. We leave this problem to future research.

# F DISCUSSION ON NON-REFERENCE METRICS

In Sec. 5.3 of our main text, we evaluate the performance of different methods on real-world data using FID score. In this section, we further consider three commonly used non-reference metrics, namely NIQE (Mittal et al., 2012), NRQM (Ma et al., 2017), and PI (Blau et al., 2018). Table 6 summarizes the detailed comparisons on WIDER-Test. Despite the strong superiority of *DifFace*, it shows a surprisingly weak performance against GAN based methods (e.g., VQFR, GFPGAN, and PSFRGAN) across these three non-reference metrics.

To explore the underlying reasons behind this phenomenon, we randomly generate 3,000 HQ face images using the pre-trained diffusion model, and then evaluate their image quality based on these non-reference metrics. The average results are denoted as "Diffusion" and marked by gray color in Table 6. One can regard this performance as the upper bound of our proposed *DifFace*.

It is interesting to observe that even this upper bound is inferior to GAN based restoration methods. Such a conclusion is unreasonably pessimistic considering the various positive results of diffusion models reported againt GANs (Song et al., 2020; Dhariwal & Nichol, 2021). More investigations are called to study this gap introduced by the aforementioned non-reference metrics.

# G INFERENCE ACCELERATION

The main limitation of *DifFace* is the slow inference speed due to the gradual sampling mechanism inherited from diffusion model. In this work, the employed diffusion model is trained with 1000 discrete diffusion steps, and the starting timestep $N$ is set as 400. To shorten the inference time, we reduce the sampling steps to 250 from 1000 for the diffusion model using the technique in Nichol & Dhariwal (2021), and the starting timestep $N$ is thus reduced to 100 from 400.

Table 7 details the performance of *DifFace* of reducing the inference process to different steps. We can see that *DifFace* performs very stably under different acceleration setup. Under the extreme setting where only 20 sampling steps is used in *DifFace*, its inference efficiency is significantly improved, having its range of running time lying between DFDNet and VQFR. Even though we speedup the inference by 20 times, *DifFace* is still superior to VQFR. Several qualitative comparisons are displayed in Fig. 13.

# H    EXTENSION TO LATENT SPACE

Instead of learning the data distribution in the image space, Rombach et al. (2022) propose LDM to model the data distribution in the low dimentional latent space of VQGAN (Esser et al., 2021), alleviating the requirement on huge computational resources. Specifically, LDM first trains a VQ-GAN model which consists of an encoder $E$ and a decoder $D$ on the training dataset $\mathcal{S} = \{\boldsymbol{x}^{(i)}\}_{i=1}^{M}$. Given any $\boldsymbol{x}^{(i)} \in \mathcal{S}$, the encoder maps it to the latent space via $\boldsymbol{z}^{(i)} = E\left(\boldsymbol{x}^{(i)}\right)$, and the decoder aims to reconstruct this image by $\boldsymbol{x}^{(i)} = D\left(\boldsymbol{z}^{(i)}\right)$. Then, a diffusion model is trained on the latent space $\mathcal{Z} = \{\boldsymbol{z}^{(i)}\}_{i=1}^{M}$ of such a VQGAN.

Extending the proposed *DifFace* to latent space can be easily achieved by moving the posterior $p(\boldsymbol{x}_0|\boldsymbol{y}_0)$ of Eq. (5) to latent space, namely

$$p(\boldsymbol{z}_0|\tilde{\boldsymbol{z}}_0) = \int p(\boldsymbol{z}_N|\tilde{\boldsymbol{z}}_0) \prod_{t=1}^{N} p_\theta(\boldsymbol{z}_{t-1}|\boldsymbol{z}_t)\mathrm{d}\boldsymbol{z}_{1:N}, \; \tilde{\boldsymbol{z}}_0 = E\left(\boldsymbol{y}_0\right), \tag{14}$$

where $\boldsymbol{y}_0$ is the degraded image, $E$ is the encoder of VQGAN. To restore the high-quality image $\boldsymbol{x}_0$ from $\boldsymbol{y}_0$, we first get a sample from $p(\boldsymbol{z}_0|\tilde{\boldsymbol{z}}_0)$, and then decode it to the image space, i.e.,

$$\boldsymbol{z}_0 \sim p(\boldsymbol{z}_0|\tilde{\boldsymbol{z}}_0), \;\; \boldsymbol{x}_0 = D\left(\boldsymbol{z}_0\right). \tag{15}$$

Similar to that in image space, the transition distribution $p(\boldsymbol{z}_N|\tilde{\boldsymbol{z}}_0)$ is also formulated as a Gaussian distribution but built in latent space:

$$p(\boldsymbol{z}_N|\tilde{\boldsymbol{z}}_0) = \mathcal{N}\left(\boldsymbol{z}_N; \sqrt{\alpha_N}f\left(\tilde{\boldsymbol{z}}_0; w\right), \sqrt{(1-\alpha_N)}\boldsymbol{I}\right). \tag{16}$$

As for the diffused estimator $f(\cdot; w)$, it is trained on the synthetic image pairs $\{\boldsymbol{x}^{(i)}, \boldsymbol{y}^{(i)}\}_{i=1}^{M}$ generated via Eq. (11) as follows:

$$\min_{w} \sum_{i=1}^{M} L\left(f\left(E(\boldsymbol{y}^{(i)}); w\right), E(\boldsymbol{x}^{(i)})\right), \tag{17}$$

where $L(\cdot, \cdot)$ is a pre-defined feature matching loss in latent space. With the formulation above, we extend *DifFace* to the latent space, and denote it as *DifFace*-Latent.

We make some initial exploration along this line by simply setting $L(\cdot, \cdot)$ of Eq. (17) as $L_2$ loss, and some visual comparisons on CelebA-Test are displayed in Fig. 14. It can be observed that *DifFace*-Latent is also able to successfully recover a realistic-looking result with clear facial details. However, its identity preservation is inferior to *DifFace*. We speculate that this drawback can be overcame by employing a more effective latent feature matching loss $L(\cdot, \cdot)$. In fact, the design of feature matching in latent space is still an open research topic, two representative works are Zhou et al. (2022) and Chen et al. (2022). We leave this to our future work.

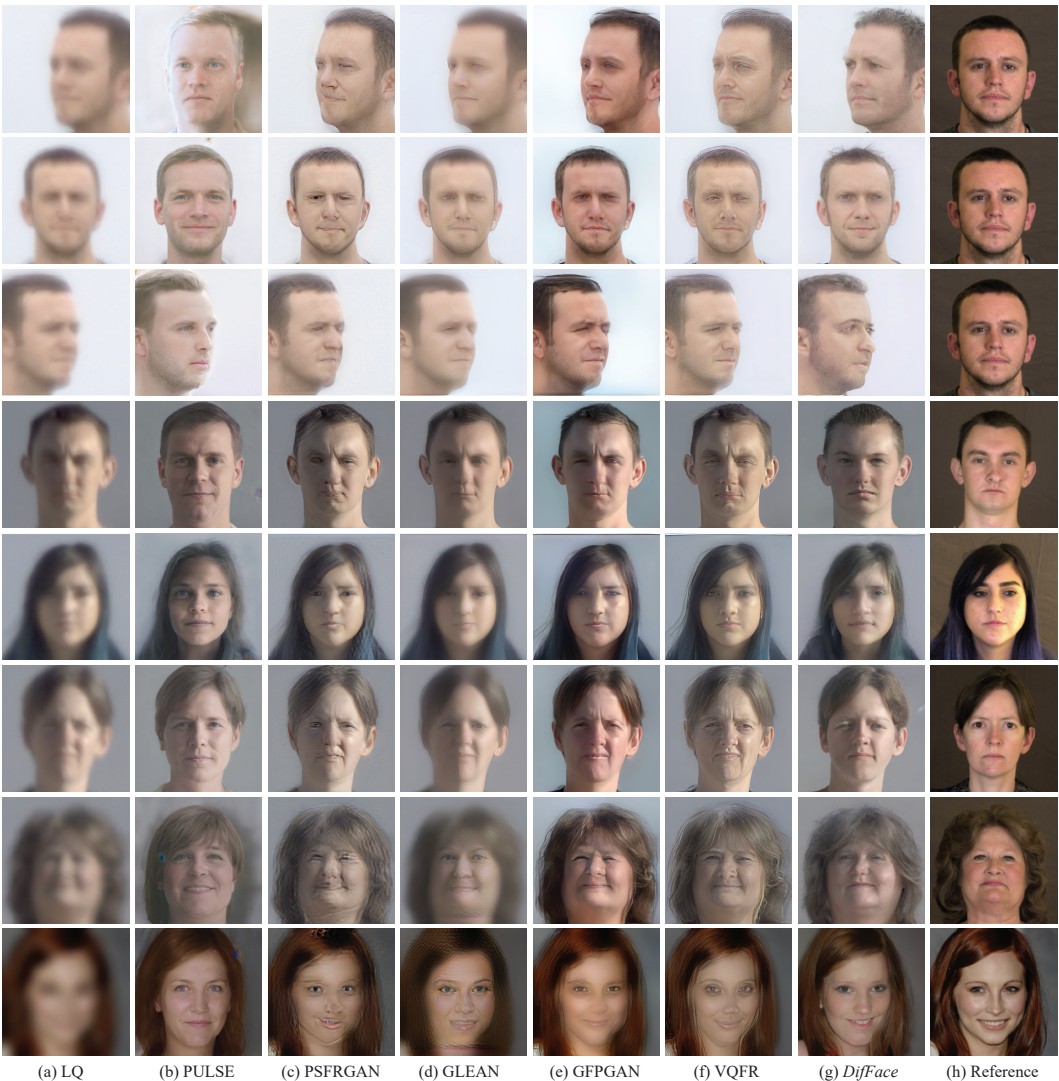

(a) LQ    (b) PULSE    (c) PSFRGAN    (d) GLEAN    (e) GFPGAN    (f) VQFR    (g) *DifFace*    (h) Reference

Figure 12: Qualitative comparisons of different methods on the real-world Turb8-Test dataset.

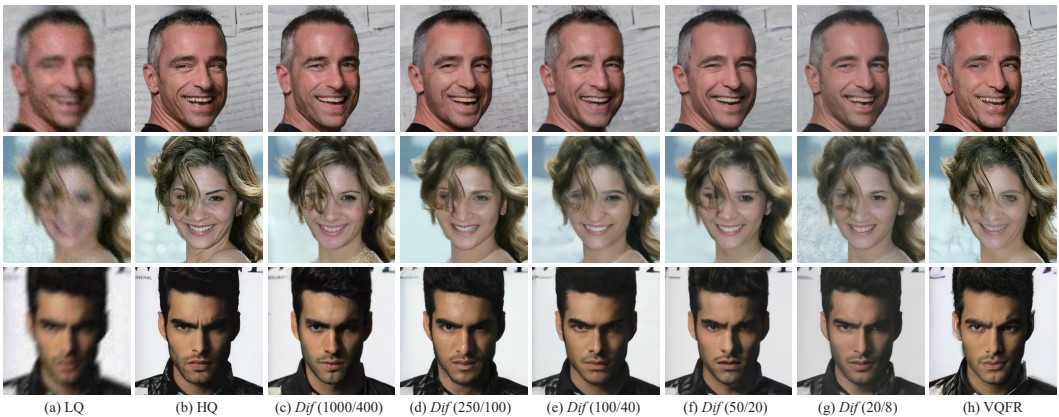

(a) LQ    (b) HQ    (c) *Dif* (1000/400)    (d) *Dif* (250/100)    (e) *Dif* (100/40)    (f) *Dif* (50/20)    (g) *Dif* (20/8)    (h) VQFR

Figure 13: Qualitative comparisons of VQFR and *DifFace* under different acceleration settings. "*Dif*(A/B)" means that the whole reverse length of the pretrained diffusion model is A, and the corresponding starting timestep $N$ in *DifFace* is set as B.

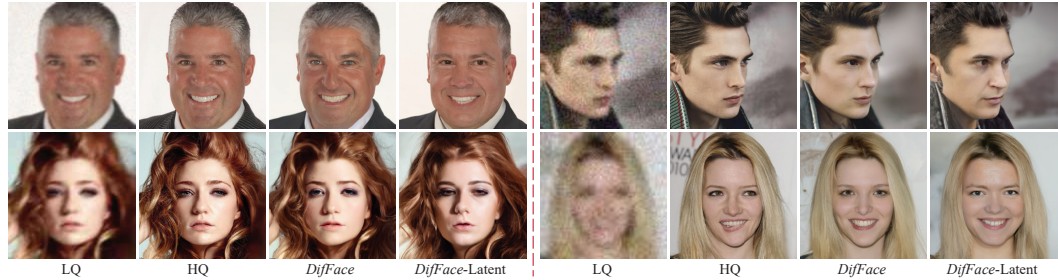

Figure 14: Qualitative comparisons of *DifFace* and its extended version (*DifFace*-Latent) in the latent space.

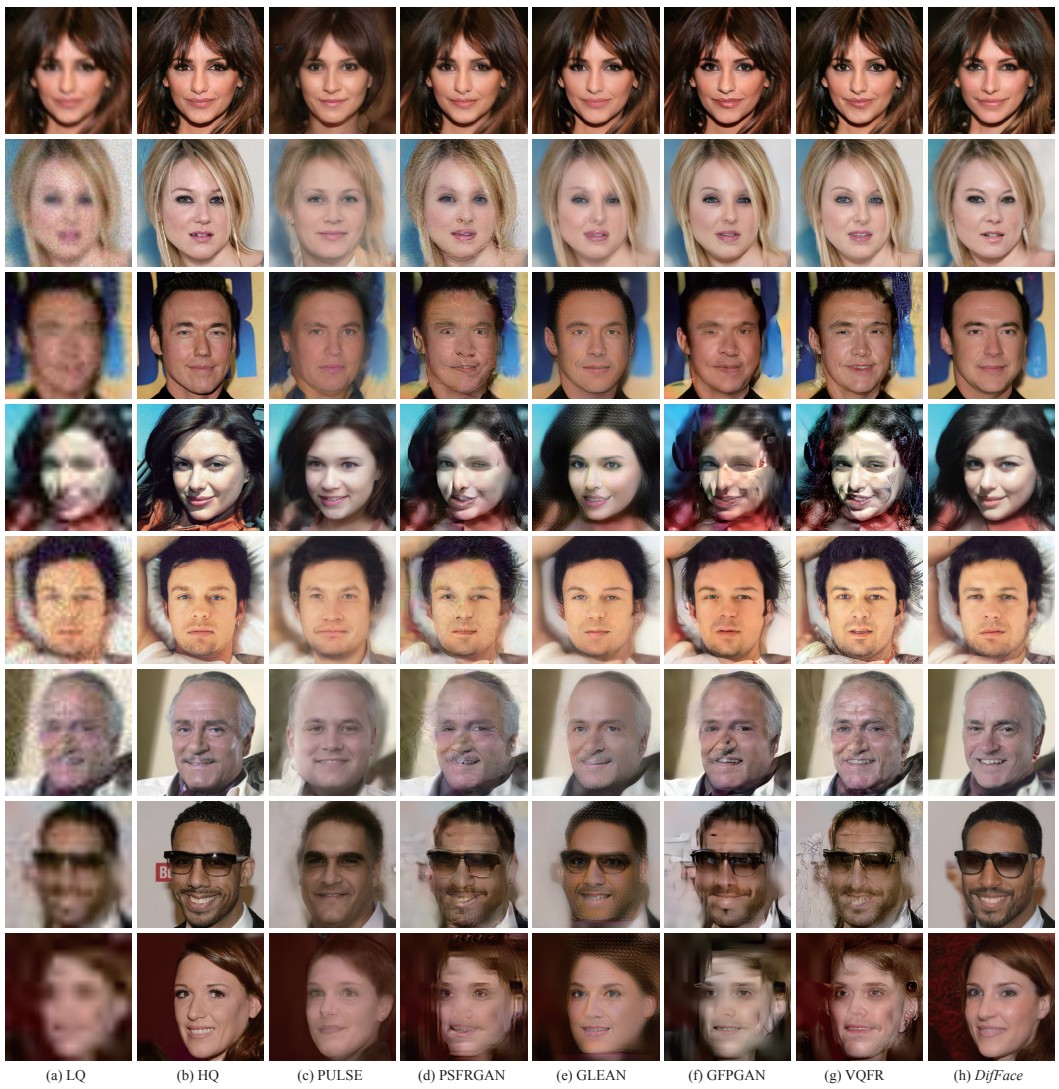

Figure 15: Visual comparisons of different method on the synthetic dataset CelebA-Test.

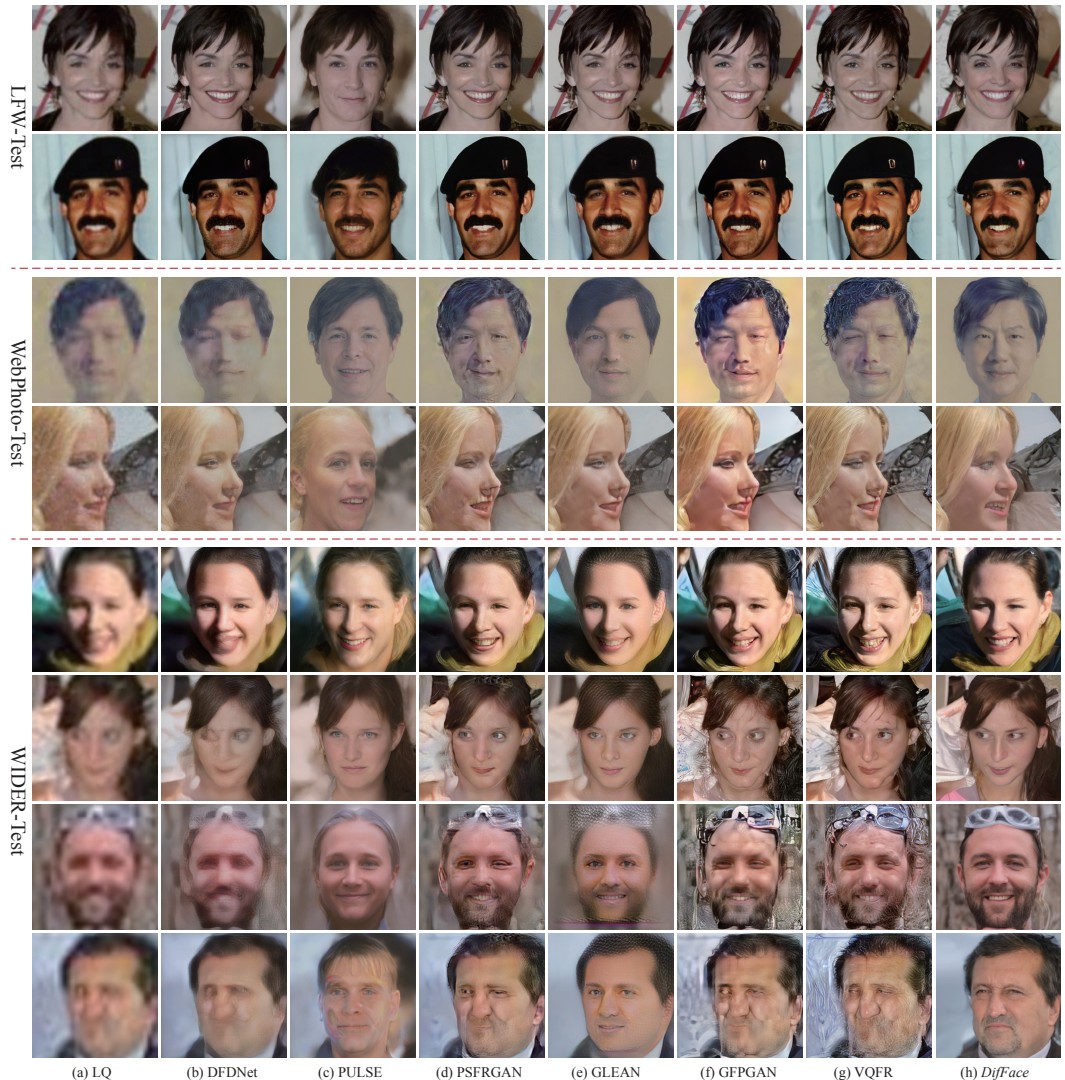

Figure 16: Visual comparisons of different method on three real-world datasets.

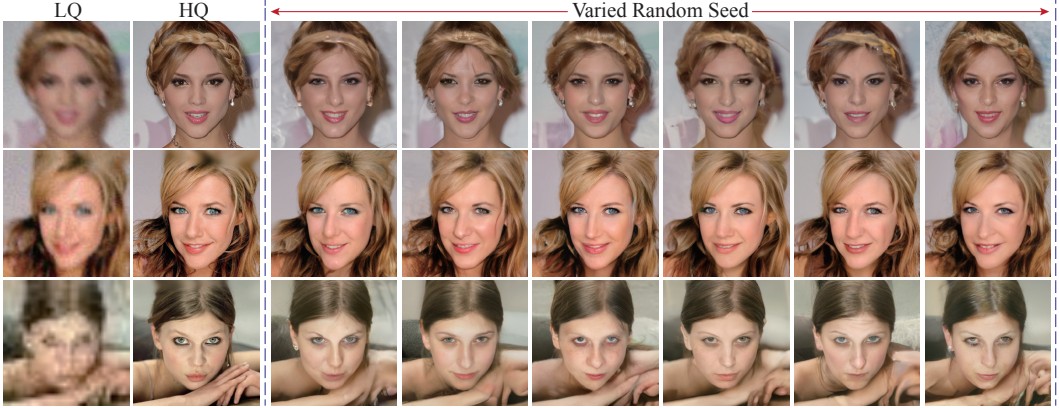

Figure 17: Three restoration examples of *DifFace* on the synthetic dataset CelebA-Test by setting different random seeds for the diffusion model.

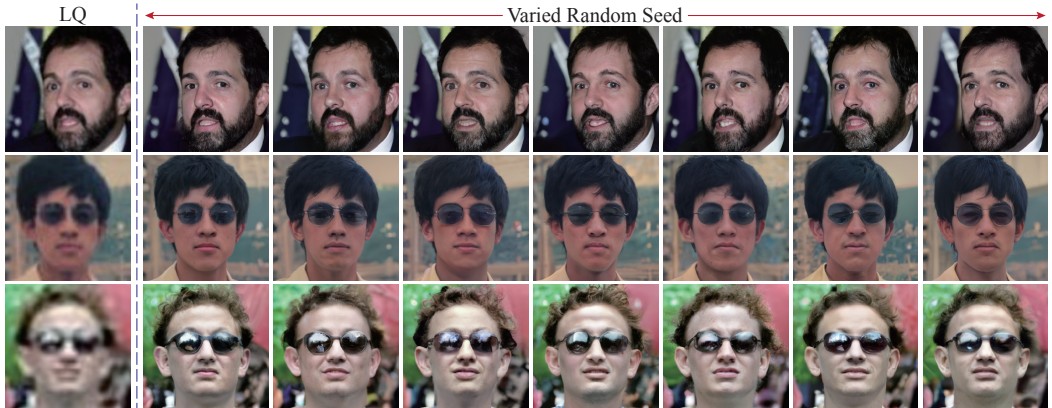

Figure 18: Three restoration examples of *DifFace* on the real-world dataset WIDER-Test by setting different random seeds for the diffusion model.

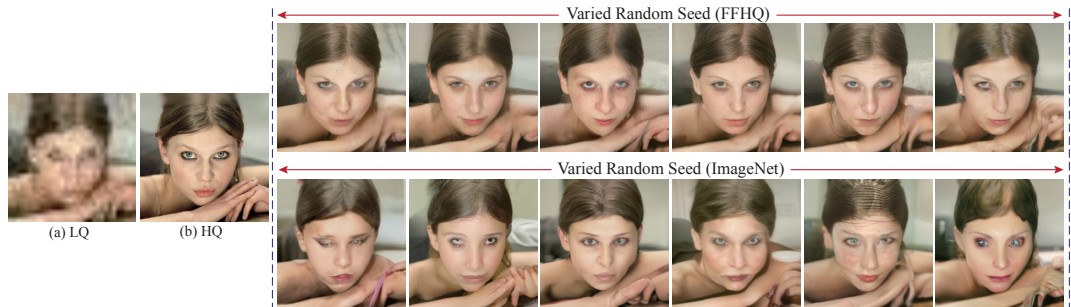

Figure 19: Qualitative comparisons of *DifFace* with pretrained diffusion models trained on different datasets. The top row employs a diffusion model trained on FFHQ (Karras et al., 2019), while the bottom row on ImageNet (Deng et al., 2009).

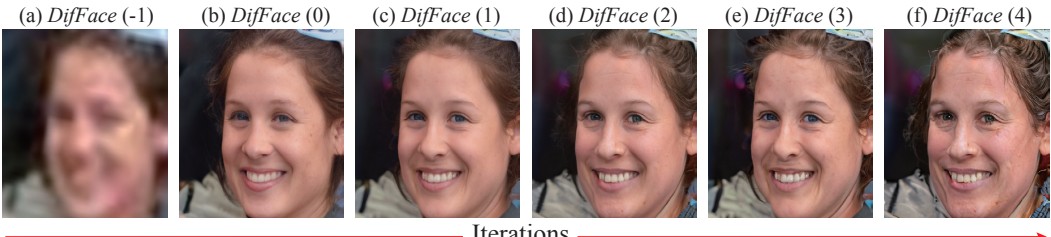

Figure 20: Visual results by iteratively applying the proposed *DifFace* on one real-world example. "*DifFace* ($i$)" denotes the restored result that takes "*DifFace* ($i$-1)" as the degraded input, and "*DifFace* (-1)" is the low-quality image.

