# OpenReview forum: "DifFace: Blind Face Restoration with Diffused Error Contraction"
_ICLR.cc/2023/Conference — Submitted to ICLR 2023_

### Official Review · Reviewer_FutU · 2022-10-15

**Confidence:** 4
**Correctness:** 4
**Technical Novelty And Significance:** 3
**Empirical Novelty And Significance:** 4
**Recommendation:** 6

**Clarity, Quality, Novelty And Reproducibility:**

The paper is well-written and easy to follow. The issue of reproducibility is not addressed in the paper. But based on its detailed discussion in Sec. 4, I tend to give authors some benefit of doubt. It should not be difficult to reproduce the results reported in this paper.

**Strength And Weaknesses:**

+ The motivation behind DifFace is clear. In view of the wide impact of probabilistic diffusion models on image synthesis, it is not surprising to see that LDM can be applied to low-level vision tasks such as blind face restoration.
+ The design of transition distribution in Sec. 3.2 and its discussion in Sec. 3.3 are easy to follow. Model analysis in Sec. 4 offers some good insight about the interpretability of model parameters (starting timestep N and diffused estimator f).
+ The reported subjective quality comparison results are convincing (e.g., Fig. 2, Fig. 7 and Fig. 8). Objective quality comparison in Table 1 also clearly favors the proposed method DifFace.

- I had thought this work will work in the latent space such as latent diffusion model (LDM, CVPR'2022). Based on the discussion in Sec. 5.4, this work has not been extended into the latent space. I am just curious about the authors' further thought about this line of extension. Maybe StyleGAN will serve as a natural candidate for the latent space?
- "All models are wrong; some are useful". It will be nice to show some failure examples for the proposed DifFace. How about its performance in the presence of occlusion or severe pose variation or atmospheric turbulence (e.g., IARPA BRIAR scenario)?
- If you supply Fig. 2(g) as the input image to DifFace, will it output an undistorted image? In other words, have you verified that the restored face image is a fixed point for the proposed restoration operator?

**Summary Of The Paper:**

This paper studies the problem of blind face restoration. To overcome the two limitations (lack of generalization property and requirement of multiple constraints), this paper proposes a new approach based on the latest diffusion model in the latent space. The key idea is to learn a posterior distribution from LQ to HQ with the help of a transition distribution from LQ to an intermediate state of pre-trained diffusion model. The design of transition distribution takes into account both the efficiency of training process and robustness to unknown degradation. Extensive experimental results are reported to demonstrate the superiority of the proposed DifFace to other competing methods.

**Summary Of The Review:**

I am mostly impressed by the reported experimental results of this paper. Its theoretical contribution is OK but incremental (the derived diffusion model still works in the pixel instead of the latent space). The computational complexity and the model parameters of this work do not compare favorably over existing methods such as GLEAN and DFDNet.

---

> ### Author Response · Authors · 2022-11-15
> **Response to Reviewer FutU**
>
> >Q1. How to extend our method to the latent space.
>
> Thanks for your interest to the extension of our method in the latent space. We have supplemented a concrete discussion on this problem in Appendix H. Please refer to our revised paper for more details.
>
> >Q2. It will be nice to show some failure examples for the proposed DifFace. How about its performance in the presence of occlusion or severe pose variation or atmospheric turbulence (e.g., IARPA BRIAR scenario)?
>
> Thanks for your suggestion. We have supplied more experiments on some degraded images by atmospheric turbulence in Appendix E, aiming to better evaluate the generalization capability of different methods on unknown degradation.
>
> We have attempted to contact with the authors of [1] to access some images of IARPA BRIAR dataset. However, due to the restriction of USA government, they cannot share this dataset with us. We thus roughly cropped eight images from this paper, each containing one low-quality image and one corresponding reference image. Figure 12 displays the qualitative comparisons of different methods on this dataset. We can see that DifFace is able to obtain photo-realistic results and evidently outperforms other methods in visual quality. In particular, the first three examples represent the same person but with large pose variations, DifFace still performs well in this scenario. Unfortunately, DifFace sometimes cannot preserve the identity very well when the input loses too much information, e.g., the last example in Fig. 12. We leave this problem to future research.
>
> >Q3. If you supply Fig. 2(g) as the input image to DifFace, will it output an undistorted image? In other words, have you verified that the restored face image is a fixed point for the proposed restoration operator?
>
> Actually, it is difficult to guarantee that the restored face image by our method is a fixed point. That’s because DifFace is indeed an iteration-based algorithm, which involves multiple sampling procedures. Each sampling will introduce some randomness as shown in Fig. 5, and inevitably leads to some changes to the identity.
>
> To empirically verify this point, we iteratively apply the proposed DifFace algorithm in the real-world example of Fig. 2 as suggested and list the results in Fig. 20. It can be observed that, with more iterations, the restored images possess clearer facial details, but with larger identity variation. This is consist with our analysis above.
>
> [1] Mei K , Patel V M . LTT-GAN: Looking Through Turbulence by Inverting GANs. ArXiv, 2021.

---

### Official Review · Reviewer_yge8 · 2022-10-24

**Confidence:** 5
**Correctness:** 3
**Technical Novelty And Significance:** 2
**Empirical Novelty And Significance:** 2
**Recommendation:** 5

**Clarity, Quality, Novelty And Reproducibility:**

The authors devise a new diffusion-based BFR approach to cope with severe and unknown degradations. The notion of formulating the posterior distribution as a Markov chain that starts from the LQ image and ends at the desirable HQ image is novel. We theoretically show that the Markov chain can compress the predicted error by a factor of less than 1.
The authors show that the image prior captured in a pretrained diffusion model can be harnessed by having the Markov chain built partially on the reverse diffusion process. Such a unique design also allows us to explicitly control the restoration’s fidelity and realism by changing the Markov chain’s length.
The authors show that BFR can be achieved without complicated losses as in existing methods. The proposed method only needs to train a neural network with the basic L2 loss, simplifying the training pipeline.

**Strength And Weaknesses:**

Pros:
1. The motivation is clear.
2. The paper is well-written and organized.
Cons:
1. More results on more challenging datasets are needed to verify the superiority of the proposed method.
2. Some related works are missing, e.g., Toward High-Quality Face-Mask Occluded Restoration, Joint Face Image Restoration and Frontalization for Recognition

**Summary Of The Paper:**

This paper proposes a new method DifFace, being able to cope with unseen and complex degradations more gracefully without complicated loss designs. Comprehensive experiments show that DifFace is better than current state-of-the-art methods, especially in cases with severe degradations.

**Summary Of The Review:**

See Strength And Weaknesses.

---

> ### Author Response · Authors · 2022-11-15
> **Response to Reviewer yge8**
>
> >Q1. More results on more challenging datasets are needed to verify the superiority of the proposed method.
>
> We employed three real-world datasets, containing different degrees of degradation, to evaluate the superiority of the proposed method in our paper. In particular, the WIDER-Test dataset comprises 970 severely degraded images from the WIDER Face dataset. To the best of our knowledge, this is the most challenging real-world dataset in the field of blind face restoration so far. In order to provide a more comprehensive comparison on this dataset, we additionally show 100 restored examples of our method and current state-of-the-art VQFR in this anonymous website (https://sites.google.com/view/wider-test/).
>
> Besides, we collect another real-world dataset denoted as Trub8-Test to evaluate the robustness of different methods. This dataset is mainly degraded by atmospheric turbulence not included in our training data. More details about this dataset can be found in Appendix E, and the qualitative comparison results on this dataset are shown in Fig. 12. We can observe that the results of the proposed DifFace are more realistic-looking compared with others, and much closer to the reference images.
>
> Therefore, it is reasonable to say that the proposed DifFace is superior to other methods, especially when dealing with severe degradations.
>
> >Q2. Missing of two related works.
>
> Thanks. We have added these two related works in Sec. 1 in our revised paper. Since these two works are with different experimental settings to ours and their source codes are not released, we thus do not compare with them.

---

### Official Review · Reviewer_SR7T · 2022-10-25

**Confidence:** 3
**Correctness:** 4
**Technical Novelty And Significance:** 4
**Empirical Novelty And Significance:** 4
**Recommendation:** 8

**Clarity, Quality, Novelty And Reproducibility:**

The paper has good quality and clarity. It is overall a good read and the work is original in my viewpoint.

**Details Of Ethics Concerns:**

The paper inherits the ethical concerns of face image generation applications. No extra concerns from my point of view.

**Strength And Weaknesses:**

Strength:

1. The paper is well written and a pleasure to read. The paper posed a very simple yet effective method for face restoration. The method is well motivated and explained, with extensive analysis of the method some design choices, e.g., choice of starting timestep N.

2. The idea is novel to me. The paper proposed a method to leverage a pre-trained diffusion model for BFR. The method is novel in the sense that it constructs a transition distribution to infer an intermediate state of the diffusion process of a high quality image from a low quality counterpart.

3. The results are generally convincing. The proposed method seems to be more robust than most previous methods that are used as baselines in the paper, with less artifact and higher fidelity in visual comparisons. Quantitative evaluations also demonstrated better performance than previous art.

Weakness:

I don't have much complaints or concerns as the author has already mentioned that one limitation of the method is that, by having to go through the diffusion process, the performance is slow.
One question that I have is the choice of the training data for the generative model. Since variants of diffusion models has been used in training image generation models for diverse images, would training on data beyond human faces provide better prior for face restoration? In Figure 13, the hand restoration is clearly failing. Is this due to the lack of training data diversity of the diffusion model since they are only trained on FFHQ?




**Summary Of The Paper:**

The paper proposed a method for Blind Face Restoration (BFR) using pre-trained diffusion models for faces. The main novelty is a design of a transition model from a low quality image that estimates an intermediate state of the diffusion process of a high quality image. This allows the method to use a pre-trained diffusion model as the image restoration back-bone. The paper is equipped with extensive analysis to motivate the approach. The paper also provided qualitative and quantitative comparisons on BFR task with several previous art. The results are generally in favor of the proposed method demonstrating robust face restoration quality.

**Summary Of The Review:**

I think the paper proposed a simple and original method for blind face restoration. The paper is well written. The experimental results are convincing. I hold a positive view of the paper and I recommend accept.

---

> ### Author Response · Authors · 2022-11-15
> **Response to Reviewer SR7T**
>
> Thanks for your positive comments to our work. We’re happy to have a further discussion with you on our work.
>
> >Q1. Choices of training data for diffusion model.
>
> I partially agree with your opinion. Restricting the training data to face images may lead to unsatisfactory results in some non-face areas, e.g., the hand in Fig. 17, but it could help to generate a more realistic-looking face image. To validate this point, we replaced the diffusion model in our method with another one trained on ImageNet. The comparative results on these two different settings are listed in Fig. 19 of our revised paper. We can see that changing the training data to ImageNet fails to improve the hand restoration and generates some distorted face images. As for the strategy of increasing the diversity of the training data, e.g., combing FFHQ and ImageNet, I am not sure how much performance gain it will bring up, since we do not  have enough computational resources to train such a huge diffusion model.
>
> I think the failure of hand restoration is mainly attributed to the severe degradation that almost destroys all the useful information in the areas of hand. It is very difficult for the diffusion model to recover such areas without extra guidance. Thus, a possible solution is to provide some additional guidance (e.g., text) for the diffusion model to address this problem.

---

### Official Review · Reviewer_sp9a · 2022-10-25

**Confidence:** 4
**Correctness:** 3
**Technical Novelty And Significance:** 2
**Empirical Novelty And Significance:** 3
**Recommendation:** 5

**Clarity, Quality, Novelty And Reproducibility:**


The paper is well-written and easy to follow. However, the novelty of the proposed method is limited.


**Strength And Weaknesses:**

Strength:
1. The idea of bringing the diffusion model into BFR task is somewhat new but the contribution of this paper is not significant.
2. The results of DifFace are better than current state-of-the-art methods and extensive experiments and analysis have been provided to evaluate the proposed method.
Weaknesses:
1. The main contribution of this paper is designing a diffused estimator to transform the degraded facial images into the latent space of diffusion model, which is not new [1] and this contribution is very limited.
2. The proposed DifFace model has the advantage of dealing with severe degradation. How about the real degradation face images with fewer degradation? Can the DifFace still restore the identity well?
3. It was claimed that this paper theoretically shows that the Markov chain can compress the predicted error by a factor of less than 1. However, no theoretical analysis can be found in this manuscript.
4. It was claimed that the proposed approach avoids using GAN and perceptual loss. However, it is not difficult to use such loss and stable results can also be obtained by those works such as GFPGAN, VQFR. In the proposed method, Eq.(9) introduce a weighted factor k_N, which strictly decreases monotonically with the timestep N. How to choose k_N and a_N? The number of inference sampling times N is also manually chosen. Compared with previous methods, the proposed method requires a more manual and empirical setting.
5. It’s still unclear why p(x_N |y_0) is a Gaussian distribution as shown in Eq.(8). How about the Laplace distribution?
6. Since the diffusion model requires a large number of sampling, i.e., 400 in this work, the inference time of DifFace is too slow compared with other methods. The number of sampling need further ablation study.

[1] Jonathan Ho, Ajay Jain, and Pieter Abbeel. Denoising diffusion probabilistic models. Advances in Neural Information Processing Systems (NeurIPS), 33:6840–6851, 2020.


**Summary Of The Paper:**

This paper proposed to use a pre-trained diffusion model as a generative prior for BFR. With a diffused estimator, the LQ images with different degradation can be transformed into the latent space of the diffusion model. A weighted L2 loss has been proposed to train the diffused estimator. Generally, the paper is well-written with good performance.

**Summary Of The Review:**

Although this paper achieves good performance, the contribution and novelty of this paper are not enough.

---

> ### Author Response · Authors · 2022-11-15
> **Response to Reviewer sp9a (Part 1/2)**
>
> > Q1. It’s still unclear why $p(x_N|y_0)$ is a Gaussian distribution as shown in Eq. (8). How about the Laplace distribution?
>
> To answer this question, it is necessary to recall our flow to design the posterior $p(x_0|y_0)$, which is the ultimate goal for blind face restoration. First, we formulate this posterior as a Markov chain (Eq. (5) of our paper), i.e.,
> $$ p(x_0|y_0)=\int p(x_N|y_0)\prod_{t=1}^Np_{\theta}(x_{t-1}|x_t)\text{d}x_{1:N}, $$
> where $p_{\theta}(x_{t-1}|x_t)$ is a pretrained diffusion model, building a bridge to leverage the image prior contained in the diffusion model. Thus, our goal turns to design a transition distribution $p(x_N|y_0)$. Second, the analysis in Sec. 3.1 provides an important observation: the marginal distribution $q(x_N|x_0)$ defined in Eq. (2) is an idle choice for $p(x_N|y_0)$. Therefore, the key principle to devise $p(x_N|y_0)$ is to approximate $q(x_N|x_0)$ using it. Third, $q(x_N|x_0)$ is a Gaussian distribution with analytical expression relying on $x_0$ as shown in Eq. (2), namely,
>
> $$q(x_N|x_0) = N(x_N;\sqrt{\alpha_N}x_0, (1-\alpha_N)I).$$
>
> This inspires us introduce a function, i.e., the diffused estimator, to estimate $x_0$, and then set $p(x_N|y_0)$ with the same Gaussian formulation as the target $q(x_N|x_0)$, i.e.,
>
> $$p(x_N|y_0)=N(x_N;\sqrt{\alpha_N}f(y_0;w), (1-\alpha_N)I).$$
>
> At this point, we have completed the design for the posterior distribution $p(x_0|y_0)$.
>
> In summary, the choice of Gaussian form for $p(x_N|y_0)$ is naturally determined by its idle target $q(x_N|x_0)$.
>
> >Q2. Theoretical analysis on the error contraction.
>
> To restore a high-quality image $x_0$ given the degraded $y_0$, we first predict an initial result  using the diffused estimator, namely $f(y_0;w)$, and denote the estimated error as $e=x_0-f(y_0;w)$. Then, we sample the latent variable $x_N$ from the designed transition distribution $p(x_N|y_0)$, i.e.,
>
> $$x_N=\sqrt{\alpha_N}f(y_0;w)+\sqrt{(1-\alpha_N)}\xi=\sqrt{\alpha_N}x_0-\sqrt{\alpha_N}e+\sqrt{(1-\alpha_N)}\xi,$$
>
> where $\xi \sim N(\xi;0, I)$. It can be seen that the predicted error $e$ is contracted by multiplying a factor of $\sqrt{\alpha_N}$ less than 1 after transformed to $x_N$. With $x_N$, we can obtain the desired high-quality image $x_0$ through reversely sampling from $p_{\theta}(x_{t-1}|x_t)$.
>
> > Q3. Remark on the $L_2$ loss to train the diffused estimator.
>
> We want to kindly give a note about the training loss for the diffused estimator. In our method, the diffused estimator can be simply trained with a vanilla $L_2$ loss, **not a manually designed weighted $L_2$ loss**, i.e.,
>
> $$ \min_w \sum_{i=1}^M \Vert f(y^{(i)};w) - x^{(i)} \Vert_2^2,$$
>
> where the image pairs $\lbrace x^{(i)}, y^{(i)}\rbrace_{i=1}^M$ are synthesized via Eq. (11). Most notably, the underlying error contraction as explained in Q2 guarantees that our method has a large error tolerance for the diffused estimator, that’s why we can simply train it using a basic $L_2$ loss.
>
> > Q4. It was claimed that the proposed approach avoids using GAN and perceptual loss. However, it is not difficult to use such loss and stable results can also be obtained by those works such as GFPGAN, VQFR. In the proposed method, Eq.(9) introduce a weighted factor $\kappa_N$, which strictly decreases monotonically with the timestep $N$. How to choose $\kappa_N$ and $\alpha_N$? The number of inference sampling times $N$ is also manually chosen. Compared with previous methods, the proposed method requires a more manual and empirical setting.
>
> Please kindly note that Eq. (9) is not our loss function to train the diffused estimator as reclaimed in Q3. **We do not need to manually choose any weighted factor in the training phase**. The only one hyper-parameter in our method is the starting timestep $N$, and we will explain how to determine it in the following statement.
>
> In fact, Eq. (9) is just used to analytically depict the discrepancy between our designed transition distribution $p(x_N|y_0)$ and its idle goal $q(x_N|x_0)$. Combined Fig. 3 and Fig. 4, such a discrepancy analysis delivers one important insight: the choice of $N$ induces a realism-fidelity trade-off for our method. This phenomenon is further illustrated by the FID (measuring realism) and LPIPS (measuring perceptual fidelity) curves in Fig. 6(a). According to the intersection of the FID and LPIPS curves, it is easy to select a reasonable timestep $N$ without any complicated empirical settings.
>
> By the way, this trade-off makes it possible to provide a human-interactive interface to explicitly control the realism and fidelity of the restored results by adjusting the parameter $N$, so as to meet different requirements in real applications.

---

> ### Author Response · Authors · 2022-11-15
> **Response to Reviewer sp9a (Part 2/2)**
>
> >Q5. Since the diffusion model requires a large number of sampling, i.e., 400 in this work, the inference time of DifFace is too slow compared with other methods. The number of sampling need further ablation study.
>
> The employed diffusion model in this work is trained with 1000 diffusion steps, and we set the starting timestep $N$ as 400 under such a setting. In practice, however, some acceleration techniques can be leveraged to largely reduce the sampling steps of the diffusion model. For example, we reduce the sampling steps from 1000 to 250 in the experiments of this paper, and thus the starting timestep $N$ in our method is also reduced to 100 from 400. Note that all the experimental results in our paper are reported under this accelerated setup. We have corrected this ambiguous writing expression in our revised paper.
>
> As claimed in the common response, we further conduct some ablation experiments to compare the performance of our method under different acceleration settings in Appendix G (see Table 7 and Fig. 13). From Table 7, we can observe that **the sampling steps of our method can be reduced to 20 steps while still having evident performance superiority to current state-of-the-art method VQFR**. Under this setting, it takes 0.92s to process an image with size 512x512, lying between DFDNet (1.40s) and VQFR (0.24s). We thus believe that the proposed method is able to meet some real-world requirements.
>
> >Q6. How about the real degradation face images with fewer degradation? Can the DifFace still restore the identity well?
>
> In fact, to comprehensively evaluate the performance of our method, the employed three real-world datasets in our paper have contained different degrees of degradation, varying from mild to severe. The images in the LFW-Test dataset are all with less degradation, and we showed three restored examples (first row of Fig. 8, first and second row of Fig.16) in our paper. Besides, three synthetic examples with ground truth images under mild degradations (first row of Fig. 7, first and second row of Fig. 15) are also displayed in our work to illustrate the capability of our method on identity preservation. It can be observed that the proposed method can also keep the identity well in cases with slight degradations.
>
> As a supplement, Fig. 6(b) and Fig. 6(c) list the performance of our method under different degrees of degradation on the synthetic dataset CelebA-Test. Our method has comparable performance with current state-of-the-art methods GFPGAN and VQFR in the scenarios with less degradation.
>
> >Q7. Differences to the work DDPM [1].
>
> We wish to clarify the difference of the proposed method to DDPM [1] and our contributions. DDPM is a generative model that learns a data distribution based on some observed data points. After training, it is able to generate new examples randomly, complying with the targeted distribution. The task of blind face restoration, however, aims at recovering the underlying high-quality image only given the degraded counterpart. The problem setting is completely different from DDPM. Hence, DDPM cannot directly deal with the task of blind face restoration.
>
> The main contribution of this work is to design a novel and theoretically sound posterior distribution for blind face restoration. The design of such a posterior distribution is not trivial, and cannot be directly accessed from DDPM. The designed posterior distribution consists of a transition distribution $p(x_N|y_0)$ and a reverse Markov chain from $x_N$ to $x_0$, as explained in Q1. The underlying insights behind such a design is two-fold. On the one hand, the devised transition distribution $p(x_N|y_0)$ induces an error contraction mechanism as shown in Q2. Thus, our method has a large error tolerance to the diffused estimator, making it possible to train the diffused estimator simply with a vanilla  $L_2$ loss. On the other hand, the partial transition from $x_N$ to $x_0$ is borrowed from a diffusion model, so as to sufficiently exploit the rich image prior knowledge contained in the pretrained diffusion model. In summary, our work can be easily distinguished from DDPM [1] either from the problem setting or the intrinsic motivation.
>
> [1] Jonathan Ho, Ajay Jain, and Pieter Abbeel. Denoising diffusion probabilistic models. Advances in Neural Information Processing Systems (NeurIPS), 33:6840–6851, 2020.

---

### Author Response · Authors · 2022-11-15
**Common Response**

We sincerely thank all reviewers for their constructive comments. As noted in Sec. 5.4, this work is mainly limited by the low inference speed. To further address this issue, we have additionally explored and analysed different acceleration setups in Appendix G, and the detailed performance comparisons are summarized in Table 7 and Fig. 13. It can be observed that the sampling steps of DifFace can be reduced to 20 steps while still having superior performance against current state-of-the-art method VQFR. Under such a setting, it takes 0.92s to process a degraded image with a resolution of 512$\times$512, lying between DFDNet (1.40s) and VQFR (0.24s). This further mitigates the limitation of DifFace on the inference speed.

---

### Decision · Program_Chairs · 2023-01-20

**Decision:**

Reject

**Justification For Why Not Higher Score:**

 The paper is a relatively straightforward application of diffusion models to image (face) restoration, and the paper focuses on the relatively narrow problem of face restoration. While the paper provides excellent for this setup, some consensus emerged that in light of serval other works studying image reconstruction with diffusion models, the contribution is somewhat limited.

**Justification For Why Not Lower Score:**

The method in the paper performs very well, and the evaluation is nearly flawless in that it considers several suitable datasets for face restoration and compares to well-chosen baselines.

**Metareview: Summary, Strengths And Weaknesses:**

The paper considers the problem of blind face restoration. The paper proposes a method called difface based on an application of diffusion models. The method performs very well for face restoration problems. The method is relatively slow owing to its dependence on diffusion models.

The paper provides an extensive evaluation, specifically, it evaluates on three real-world datasets and compares the performance to several reasonable baseline methods.

Three critiques that remain are the following. Those critiques and the paper were discussed amongst the AC and the reviewers during a virtual meeting and through an online discussion.
- The paper is a relatively straightforward application of diffusion models to image (face) restoration, and the paper focuses on the relatively narrow problem of face restoration. While the paper provides excellent for this setup, some consensus emerged that in light of serval other works studying image reconstruction with diffusion models, the contribution is somewhat limited.
- The paper writes 'We theoretically show that the Markov chain can compress the predicted error by a factor of less than 1'. It was unclear to several reviewers where this statement comes from, and this is somewhat important, since the authors write that 'the main contribution of this work is to design a novel and theoretically sound posterior distribution for blind face restoration.', so I and the reviewer expected a concrete analytical statement, which is not provided. The authors explained on openreview where this statement emerges, but the paper still doesn't contain an explanation. Also, the error contraction is more an observation based on one equation rather than a proof or a statement, which the abstract suggests.
- No supplement to re-produce the results is provided.


**Summary Of Ac-Reviewer Meeting:**

In a virtual meeting, we discussed the issues that are already mentioned in the reviews, such as no support for the theoretical claims made in the paper (such as the Markov chain being smaller than one, and on the length of the Markov chain). The reviewers mentioned that the experimental comparisons in the paper are sufficient but that the paper is a relatively straightforward application of diffusion models, well executed, but not particularly novel.

Here is a summary of the written reviews:

R1 notes that DifFace provides state-of-the-art performance and an extensive evaluation has been carried out to evaluate the method.
However, the reviewer notes that the idea of using diffusion models for face restoration is not new and refers to the paper [1] ``Denoising Diffusion Probabilistic Models''. R1 finds the contribution not sufficient.
The paper [1] doesn't study face reconstruction, but a variety of other papers are studying image reconstruction with diffusion models. We discussed online whether in the light of this, the contribution is sufficient.

R1 also noted that ``It was claimed that this paper theoretically shows that the Markov chain can compress the predicted error by a factor of less than 1. However, no theoretical analysis can be found in this manuscript.'' The authors explain in the response how this statement follows directly from one line in their manuscript.

R2 (8) finds the paper to be well written, the idea to be novel, and the result convincing.

R3 (5). Finds that the motivation is clear and the paper is well-written and organized. However, the reviewer notes that `more results on challenging datasets' are needed. The authors respond that they are alreadly evaluating on three real-world datasets, which they indeed do. I don't think that is an issue.

R4 (6). Finds that the motivation is clear, the paper well written, the results are convincing, and the experimental evaluation is sound. However, the theoretical and methodological contribution is incremental.